# The Liverworts of the Murmansk Region (North-West Russia): Providing an Annotated Checklist as a Basis for the Monitoring and Further Study of Liverwort Flora

**DOI:** 10.3390/plants14111590

**Published:** 2025-05-23

**Authors:** Nadezhda A. Konstantinova, Evgeny A. Borovichev, Anna A. Vilnet

**Affiliations:** Avrorin Polar-Alpine Botanical Garden-Institute, Kola Science Center, Russian Academy of Sciences, Apatity 184209, Russia; e.borovichev@ksc.ru (E.A.B.); anya_v@list.ru (A.A.V.)

**Keywords:** Hepaticae, Marchantiophyta, taxonomic diversity, distribution, red-listed species, Europe

## Abstract

An annotated list of liverworts of the Murmansk Region is compiled based on a critical compilation of publications and label data available in the information system CRIS (L.). It includes 210 species, 2 subspecies and 8 varieties, which is 59 species more than in the list of species published in 1982. Ten taxa were excluded through comparison with the previous list and later publications, due to misidentifications or new taxonomical treatments. Annotations for each species include the synonyms under which they were listed for the region; the category of threat in the Red Data Books of Europe, Russia and the Murmansk Region; links to the most representative publications on occurrence in each of nine accepted biogeographic provinces of the region; and at least one specimen number of the KPABG or INEP herbaria in the case of the absence of published data. In total, we provide 259 new records for different provinces based on herbaria KPABG (205 records) and INEP (52 records). Additionally, there are links to publications on the nucleotide sequence data of 149 specimens obtained for 82 species and for 1 variety from the Murmansk Region, including 14 specimens (11 species), published here for the first time. Species threatened in Europe, Russia and the Murmansk Region are discussed and future perspectives of liverwort study in the Murmansk Region are outlined.

## 1. Introduction

The first annotated list of liverworts of the Murmansk Region was published in 1982 [1]. Over the past 40 years, ideas about the taxonomy and the nomenclature of species have changed significantly and therefore the list of liverworts of the Murmansk Region needed to be revised. In addition, during a fairly active study of the diversity of liverworts in the region, new species for the region and new localities of species previously known by single records were found. The main purpose of this work is to summarize the results obtained during more than four decades of liverwort studies in the Murmansk Region and to compile an up-to-date annotated list of species confirmed by modern research.

### 1.1. A Brief History of the Study of Liverworts in the Murmansk Region

The first studies of bryophytes in the territory of the Murmansk Region were apparently conducted by the Swedish bryologist J. Ångström in 1843; its results were included in the summary of [2]. Subsequent studies by Scandinavian scientists, the most significant of which were the expeditions undertaken by V. Brotherus in 1872, 1885 and 1887, included studies of the Rybachij Peninsula, the Murmansk coast, the eastern and southern coasts of the Kola Peninsula, and the Khibiny Mts., Chuna-tundra Mts. and mountains in the vicinity of Kandalaksha City. The mosses collected in these expeditions, along with specimens collected in 1889 by A.O. Chilman in the central parts of the Murmansk Region, mainly in the Khibiny and Lovozero Mts., as well as specimens collected by K. Regel, who worked in the region in 1911–1913, were identified by Brotherus and published [3,4]. Liverworts collected in these expeditions, stored mainly in the herbarium of Helsinki, were identified by various bryologists (H. Buch, S.O. Lindberg, etc.); however, reports similar to those of Brotherus have not been published and information is scattered across various publications, including, for example, the books of H. Buch, H.W. Arnell and S. W. Arnell [5,6,7].

In the first half of the 20th century, L.I. Savicz-Lyubitskaya (Savicz) was engaged in collecting liverworts in the region, but many of her collections have not been identified. Moreover, for the manual of liverworts of the north of the European part of the USSR [8], authors used literary data, and records of species for the Murmansk Region in this book were based on the publication of H.W. Arnell [6]. Geobotanists also collected liverworts when describing vegetation, since liverworts sometimes make up a significant proportion of the vegetation cover of the Murmansk Region. However, only widespread species have been recorded in such works.

By the middle of the 20th century, the most well-studied area in relation to liverworts was the south-west of the region, which previously belonged to the Finnish province of Kuusamo and, in particular, the Kutsa National Park, for which 94 species were cited [9]. Another publication [10] provides an annotated list of species for the Rybachij and Srednij Peninsulas situated in the most north-western part of the Murmansk Region, previously belonging to Finland, the so-called Lapponia Petsamoënsis of Scandinavian authors. The information on liverworts of the Murmansk Region accumulated up to the middle of the 20th century is summarized by S. W. Arnell [7]. He cites the distribution of species across biogeographic provinces in Scandinavia, including six provinces recently situated in the Murmansk Region: Lp (Ponoin Lappi), Lm (Murmannin Lappi), Lim (Imandran Lappi), Lps (Petsamon Lappi), Lt (Tuloman Lappi), and Lv (Varsuga Lappi). A little later, a very incomplete bryophyte list of the Lapland State Nature Reserve was published by N.M. Pushkina [11].

Since the mid-1960s, R. N. Schljakov has studied the liverworts of the Murmansk Region as part of the study of liverworts of the north of Russia. He has collected many specimens in the region, which have formed the basis of the herbarium of liverworts of the Polar-Alpine Botanical Garden-Institute (KPABG). In several expeditions to remote and hard-to-reach regions of the Murmansk Region, Schljakov, as well as the lichenologist A. V. Dombrovskaya and botanist M. L. Ramenskaya, collected hundreds of specimens, many of which were identified by Schljakov and incorporated into the KPABG herbarium. In particular, many specimens were collected in the valley of the Iokanga River (1965), in the lower reaches and valley of the Lower Ponoy and Rusinga Rivers, Orlov Cape (1968, 1972), and in the middle reaches of the Ponoy River (1968). At the same time, Schljakov studied the morphology and taxonomy of some complex groups of liverworts—for example, the genera Lophozia, Scapania—leading to the description of new species and varieties. Some of these descriptions are based on specimens from the Murmansk Region (see below).

Since 1973, N. A. Konstantinova has participated in the study of liverworts in the Murmansk Region, starting with the study of the Khibiny Mts. liverwort flora [12,13,14]. Then, together with Schljakov, she collected liverworts in some hard-to-reach and remote areas of the Murmansk Region such as the lower reaches of the Teriberka River (1977), Kildin Island (1977), Kovdozero Lake Basin and the vicinity of the village Kovdor (1977) and the Rybachiy Peninsula (1978, 1981). The specimens collected during the expeditions were partially identified and incorporated into the KPABG herbarium, and the results were published [1,14,15].

The next stage was the study of the liverwort flora of the Lovozero Mts. Bryophytes in these mountains were collected by O. A. Belkina and A. Yu. Likhachev from 1982 to 1985, and by N. A. Konstantinova in 1984. The results of the study of liverworts collected in the Lovozero Mts. have been published in a series of papers and are summarized in [16,17].

Further work by Konstantinova aimed at studying the liverworts of previously unexplored or poorly studied areas: the Kandalakshskie and Kolvitskie Mts. (1985), the south-west part of the Murmansk Region (1986), the Lavna Tundra Mt. (1987); the Chiltald Mts. (1988) and the Jon-N`yugoajv Mts. (1989). In each of these mountain ranges, the work was carried out for about 2–3 weeks and several hundred liverwort specimens were collected. As a result, the distribution in the Murmansk Region of many species has been greatly clarified, and it has been shown that many species previously known from single localities are not rare and are even widespread and abundant in suitable localities [18].

From 1988 to 1991, the study of liverworts was focused on the flora of liverworts in the Kandalaksha State Nature Reserve, summarized, along with earlier publications, in two main works [19,20]. The expedition to Keivy Range in 1997 is worth mentioning, with collection taking place along the route of the all-terrain vehicle from Kirovsk to Oktyabrsky to the final point in the Iokanga River Valley. The specimens collected in this expedition were identified, but not published. The same is true for the collection which took place during the survey of the Gremyakha-Vyrmis deposit, studied in 1999.

While working on his PhD thesis, “*Monographic study of the genus Lophozia*”, V. A. Bakalin collected liverworts, including in some areas of the Murmansk Region. He clarified the distribution of species of the genus in the region and replenished the herbarium of KPABG with many specimens of liverworts [21,22].

Since 2004, E. A. Borovichev has studied liverworts of large mountain ranges of the Lapland State Nature Reserve: the Salnye Tundra Mts. [23,24], Chuna-tundra Mts., Monche-tundra Mts [25] and Nyavka-tundra Mts. [26]. Summarized annotated lists with an analysis of the liverwort flora of the entire reserve were later published [27,28]. Then, Borovichev studied the diversity of liverworts in a very small area (total 147.3 km^2^) in the Pasvik State Nature Reserve, located in the extreme northwest of the Murmansk Region [29,30].

In recent years, the focus has been on studying the diversity of liverworts in existing and planned specially protected natural areas [31,32,33,34,35], on clarifying the distribution and ecology of rare and threatened species [36,37,38,39,40,41,42,43], and also on the study of liverworts of territories located in industrial pollution zones [44]. The main results of the study of liverworts of the Green Belt of Fennoscandia (territories along the border with Finland and Norway) within the Murmansk Region have been summarized [45,46].

A notable landmark in the study of the diversity of liverworts in the Murmansk Region was the publication and distribution of “Bryophyta Murmanica Exsiccata”. A total of three issues were published [47,48,49]. They provide labels of 53 specimens, including 50 species of liverworts. Later, the publication of specimens collected in the Murmansk Region was continued in a series of “Hepaticae Rossicae Exsiccatae” [50,51,52,53], where 17 specimens of 13 previously unpublished for Murmansk Region species have been published.

### 1.2. Study Area

The Murmansk Region is located in the north-west of Eurasia and occupies the Kola Peninsula and part of the adjacent Scandinavian Peninsula. In terms of its area (144,000 square kilometers), it is slightly larger than, e.g., Greece and Bulgaria and about 2.4–2.7 times smaller than Finland (337,030) or Norway (385,186), bordering it. The proximity and influence of the warm current of the Gulf Stream causes a relatively mild suboceanic climate—unusually warm for such a latitude—and snowy winters and cool rainy summers. The average temperature of the coldest winter months (January, February) does not fall below −13° Celsius in the center of the region, −9° on the coast of the Barents Sea, and −11° on the coast of the White Sea, and the average temperature in July ranges from 10° to 14 °C in the center of the region and from 9° to 11° along the sea coast. Annual precipitation reaches 1000 mm or more in the mountains, 600–700 mm on the Murmansk coast and 500–600 mm in other areas [54,55]. The average air temperature over the past 20–30 years in all seasons of the year is the highest compared to similar periods since 1802 [56,57,58]. Modern warming in the region has been most pronounced since the 1990s and continues to this day, although temperature changes do not have a permanent feature: periods of warming are replaced by periods of cooling [58].

A wide variety of landscape forms—relatively high (up to 1200 m above sea level) mountains with three vegetation belts; rocky placers; fast mountain streams interspersed with vast plains, partly occupied by large swamp areas, especially in the east peninsulas, with a well-developed river and lake network; and noticeable anthropogenically disturbed territories with a network of country roads, and with disturbed vegetation around mines and factories—determine the wide variety of habitats for bryophytes in general and for liverworts in particular. Most of the territory of the region is located in the subzone of the northern taiga, and only a small strip in the north and east along the coast of the Barents and White Seas belongs to the subzone of the southern tundra, separated from the northern taiga by a narrow strip of forest tundra and birch forests (Figure 1 and Figure 2). In the highest mountains (Khibiny and Lovozero Mts., Chuna-tundra, Monche-tundra, Chiltald, Lavna-tundra), the upper vegetation belts are characterized by nival vegetation, screes and boulder fields.

## 2. Results

The annotated list of liverworts includes 210 species, 2 subspecies and 8 varieties. The nomenclature follows [59], except for some names, which have been published later, in particular *Blepharostoma brevirete, Protochilopsis grandiretis*.

The species in the list are arranged in alphabetical order. Common synonyms and those that are given in previous publications on the Murmansk Region, including the beginning and middle of the 20th century, are provided in square brackets. After the name of the species, we provide the frequency of occurrence of the species in the region, which was determined by the number of documented herbarium specimens using the following scale: rare species that have 1 to 5 documented localities; sporadic species with 6 to 20 known localities, frequent species that have 21–40 documented localities, and common species that have more than 40 documented localities. This is followed by the threatened status of species in Europe in accordance with the European Red list [60], the status in the Red Data Book of Russia [61] and then the Red Data Book of Murmansk Region [62]. The abbreviations of the threatened status in Europe (EU) are the same as those adopted in Europe [60], as follows: CR—critically endangered; EN—endangered; VU—vulnerable; NT—near threatened; LT—least concern; DD—data deficient; NE—not evaluated. The abbreviations of the threatened status in the Red Data Book of Murmansk Region (MR) [62] are as follows: 1a: (CR)—critically endangered; 2: (VU)—vulnerable; 3: (NT)—rare, near threatened; 4: (DD)—data deficient; 5: special status. The abbreviations of the threatened status in the Red Data Book of Russian Federation (RF) [61] are as follows: 2: (VU)—vulnerable; 3: (NT)—rare, near threatened. This is followed by citing publications with general information about the distribution and ecology of species in the Murmansk Region and then publications with molecular data for specimens from the Murmansk Region; the latter are given in bold and italics. The specimens that were previously deposited and published in GenBank, but which have not been cited in any papers, are provided with the herbarium number in the braces; specimens that are deposited in GenBank for this paper are provided with herbarium and accession numbers in the braces in italic and bold. The publications with specific data on the localities or distribution of the species are provided by the nine biogeographic provinces accepted here according to [1] and (Figure 1) and are given in the order of earlier to later. If there are no published data for a certain biogeographic province, but there are identified specimens of the species in the herbaria (KPABG or INEP), then we provide a link to at least one specimen cited in the open information system L. (CRIS) [63], confirming the occurrence in this particular province, indicating the collector and the herbarium specimens number in square brackets. The numbers of the biogeographical provinces are indicated by Roman numerals in bold. The publications with the first record of the species for the Murmansk Region are given in bold. The numbers of the relevant comments are shown in curly brackets at the end of the description of distribution.

### 2.1. Annotated List of Liverworts

*Aneura mirabilis* (Malmb.) Wickett & Goffinet [*Cryptothallus mirabilis* Malmb.]. Rare EU: NT; RF: 3 (VU); MR: 3 (NT); **VI:** [18,64], Khibiny Mts., Wiehle [KPABG(H) 4169].

*Aneura pinguis* (L.) Dumort. [*Riccardia pinguis* (L.) Gray]. Common. EU: LC; [1,7]; **I:** [10,15,65]; **II:** Teriberka River Basin, Konstantinova [KPABG(H) 6072]; **III:** Rusinga River, Borovichev [INEP152561]; **IV**: [30]; **V:** [23,24,25,26,27,44]; **VI**: [12,13,17,33,35,66]; **VII:** [9,67]; **VIII**: [19,20]; **IX**: [19,20].

*Anthelia julacea* (L.) Dumort. Sporadic. EU: LC; [1,7]; **I**: [10,15]; **II:** [7]; **IV**: [30]; **V**: [24,25,27]; **VI**: [12,17,66]; **VII:** [9,67].

*Anthelia juratzkana* (Limpr.) Trevis. Common. EU: LC; [1]; **I:** near Polarny Town, **Savicz-Lyubitskaya [KPABG(H) 2248]**; [10,15]; **II:** Kildin Isl., Konstantinova [KPABG(H) 2247]; **III:** Rusinga River, Borovichev [INEP152587]; **IV**: [30]; **V**: [24,25,26,27,44,68] **VI**: [12,17,33,66]; **VII:** [9,67]; **VIII:** Kandalakshskie Mts., Likhachev [KPABG(H) 4037]; **IX:** Mikkov Isl., Borovichev [INEP153674].

*Apopellia alpicola* (R.M. Schust. ex L. Söderstr., A. Hagborg & von Konrat) Nebel & D. Quandt [*Pellia alpicola* R.M. Schust. ex L. Söderstr., A. Hagborg & von Konrat]. Rare. MR: 3 (NT) (as *Pellia endiviifolia* (Dicks.) Dumort.) Rare. EU: NE; **VIII**: [69] {***KPABG(H)1010***}, [19,70] *(as Pellia endiviifolia*).

*Apopellia megaspora* (R.M.Schust.) Nebel & D.Quandt [*Pellia megaspora* R.M.Schust.]. Rare. EU: NE; **VII**: [71] {***KPABG(H) 2081***}.

*Arnellia fennica* (Gottsche) Lindb. Rare. EU: LC; MR: 3 (NT); [1]; **III:** [36,72]; **V**: [23,25,26,27,73]; **VII:** [9,46,67,74].

*Asterella lindenbergiana* (Corda ex Nees) Lindb. ex Arnell. Rare. EU: LC; MR: 2 (VU); **VIII:** Voron`i Tundra, Leshaya Mt. [75] [***KPABG(H) 21364***].

*Barbilophozia barbata* (Schmidel ex Schreb.) Loeske [*Lophozia barbata* (Schmidel ex Schreb.) Dumort.]. Sporadic. EU: LC; [1,7]; [76] {***KPABG(H) 697, 8744, 12782, 14120***}; **I**: [7]; **II:** Teriberka Settlement, Borovichev [INEP 152657]; **III:** near Ponoj Village, Dombrovskaya [KPABG(H) 18334]; **IV**: [30]; **V**: [24,25,26,27,44]; **VI**: [12,17,66]; **VII**: [9,67]; **VIII**: [19,20]; **IX**: [19,20].

*Barbilophozia hatcheri* (A. Evans) Loeske [Lophozia hatcheri (A. Evans) Steph., Barbilophozia lycopodioides var. hatcheri (A. Evans) Schljakov]. Common. EU: LC; [1]; [76] {***KPABG (H)716, 2832, 2852, 2931, 3355, 9491, 9609***}; **I:** [10,77]; **II:** Iokan’ga River Basin, Sukhaya River Valley, Schljakov [KPABG(H) 14524]; **III**: Lower Ponoj, near Ponoj Village [KPABG(H) 3513]; **IV**: [30]; **V:** [24,25,26,27,44]; **VI:** [33,66]; **VII:** [9,67]; **VIII**: [19,20]; **IX**: [19,20].

*Barbilophozia lycopodioides* (Wallr.) Loeske [*Lophozia lycopodioides* (Wallr.) Cogn.]. Common. EU: LC; [1]; [76] {***KPABG(H) 2846, 2931, 20178***}; **I:** [15]; **II:** Teriberka Settlement, Borovichev [INEP 152675]; **III:** Rusinga River, Borovichev [INEP152577]; **IV**: [30]; **V**: [24,25,26,27,44]; **VI**: [12,17,33,35,66]; **VII:** [9]; Konstantinova [KPABG(H) 2723]; **VIII:** [19,20]; **IX**: [19,20].

*Barbilophozia rubescens* (R. M. Schust. & Damsh.) Kartt. & L. Söderstr. [*Lophozia rubescens* R. M. Schust. & Damsh.]. Sporadic. EU: DD; MR: 3 (NT); [76] {***KPABG(H) 520, 699, 2465, 2505, 2655, 2718, 2754, 11233, 2017***}; **IV**: [30,46]; **V**: [26,27,72]; **VIII:** [19]; **IX**: [19].

*Barbilophozia sudetica* (Nees ex Huebener) L.Söderstr., De Roo & Hedd. [*Lophozia alpestris* auct. non (Schleich. ex F.Weber) A.Evans, *Lophozia sudetica* (Nees ex Huebener) Grolle, *Pseudolophozia sudetica* (Nees ex Huebener) Konstant. & Vilnet,]. Common. EU: LC; [1,7,22]; [76] {***KPABG(H) 9640, 12772***}; **I:** [15,22]; **II:** [22]; **III:** [22]; **IV**: [30]; **V:** [24,25,27,28]; **VI**: [12,17,33,66]; **VII:** [9]; **VIII:** [22]; **IX**: [19,22,49].

*Barbilophozia sudetica var. anomala* (Schljakov) Konstant. var. nova [*Lophozia sudetica* var. *anomala* (Schljakov) Schljakov; *Lophozia alpestris* (Schleich. ex F.Weber) A. Evans var. *anomala* Schljakov, *Lophozia sudetica* (Nees ex Huebener) Grolle var. *anomala* (Schljakov) Schljakov, *Lophozia rufescens* Schljakov, *Pseudolophozia sudetica* (Nees ex Huebener) Konstant. et Vilnet var. *anomala* (Schljakov) Konstant. et Vilnet]. Sporadic. [22,78,79]; **II:** [22]; **III:** [22]; **IV:** [22,66]; **V:** [22]; **VI:** [18] (as *Lophozia rufescens*); [22]; **VII:** [22]. {1}

*Blasia pusilla* L. Sporadic. EU: LC; [1,7]; **I**: [65,77]; **II:** Teriberka Settlement, Borovichev [INEP 152661]; **III:** Rusinga River, Borovichev [INEP152565]; **IV**: [30]; **V**: [24,25,26,27,44]; **VI**: 12, 13, 17, 66]; **VII:** [9,67]; **VIII**: [19,20]; **IX:** Kanda River Valley, Konstantinova [KPABG (H) 2113].

*Blepharostoma brevirete* (Bryhn et Kaal.) Vilnet & Bakalin [*Blepharostoma trichophyllum* (L.) Dumort. subsp. *brevirete* (Bryhn & Kaal.) R.M.Schust., *Blepharostoma trichophyllum* var. *brevirete* Bryhn & Kaal.]. Sporadic. EU: LC; [1,80]; {***KPABG(H) 123086***}; **I**: [77]; **V**: [25,26,27]; **VI**: [17]; **VII:** [9]; **VIII**: [19]. {2}

*Blepharostoma neglectum* Vilnet et Bakalin. Rare. EU: NE; **VIII:** [80], Umba River Valley, Konstantinova {***KPABG(H)12268***}. {2}

*Blepharostoma primum* Vilnet et Bakalin. Rare. EU: NE; **I:** [80], Ainovy Islands, Belkina {***KPABG(H) 20464***}. {2}

*Blepharostoma trichophyllum* (L.) Dumort. s.lat. Common. EU: LC; [1]; [80] {***KPABG(H) 20760***}; **I:** [15]; **III:** Rusinga River, Borovichev [INEP152554]; **IV**: [30]; **V**: [24,25,26,27]; **VI**: [12,17,66]; **VII:** [9,67]; **VIII:** [19,49]; **IX:** [19,49].

*Calycularia laxa* Lindb. & Arnell. Rare. EU: CR; MR: 2 (VU); **I**: [73]; **III**: [36,73]; **IV**: [30].

*Calypogeia integristipula* Steph. [*Calypogeia meylanii* H.Buch, *Calypogeia neesiana* var. *meylanii* (H.Buch.) R.M.Schust., *Calypogeia neesiana* auct. p.p. non (C.Massal. & Carestia) Müll.Frib.]. Frequent. EU: LC; [1]; **I:** [15]; **II:** Teriberka Settlement, Borovichev [INEP 152673]; **III:** Lower Ponoj River, near Ponoj Village, Schljakov [KPABG (H) 5127]; **IV**: [30]; **V**: [24,25,26,27]; **VI**: [12,17,33,66]; **VII:** [9,67]; **VIII:** [19,20]; **IX:** [19,20].

*Calypogeia muelleriana* (Schiffn.) Müll.Frib. Frequent. EU: LC; [1]; **I:** [15]; **III:** Schljakov, Rusinga [KPABG(H) 2602]; **IV:** [30]; **V**: [24,26,27,28]; **VI**: [12,17,33,66]; **VII:** Ulvinen [KPABG(H) 6861]; **VIII**: [19]; **IX**: [19]. {3}

*Calypogeia neesiana* (C.Massal. & Carestia) Müll.Frib. Frequent. EU: LC; [1]; **I:** [15,65]; **II:** Iokan’ga River basin, Konstantinova, [KPABG(H) 6215]; **III:** Lower Ponoj River, near Ponoj Village, Rusinga River, Borovichev [INEP152588]; **IV**: [30]; **V**: [24,25,26,27]; **VI**: [12,17,66]; **VII:** [9,67]; **VIII**: [19]; **IX**: [19].

*Calypogeia sphagnicola* (Arnell & J.Perss.) Warnst. & Loeske [*Calypogeia muelleriana* f. *sphagnicola* (Arnell & J.Perss.) Schljakov]. Frequent. EU: LC; **II:** Teriberka Settlement, Borovichev [INEP 152650]; **III:** Rusinga River, Borovichev [INEP152582]; **IV**: [30]; **V**: [24,25,26,27]; **VI**: [12,17,66]; **VII:** [9,67]; **VIII**: [19,20]; **IX**: [19,20].

*Calypogeia suecica* (Arnell et J. Perss.) Müll. Frib. Sporadic. EU: LC; MR: 3 (NT); **V**: [23,24,26,27]; **VIII:** [81].

*Cephalozia ambigua* C.Massal. [*Cephalozia bicuspidata* subsp. *ambigua* (C. Massal.) R.M. Schust.]. Frequent. EU: LC; [1,7]; [82] {***KPABG(H) 124264; 5252: PQ605327; 5256: PQ605326***}; **I:** [15]; **II**: Teriberka River Basin, Schljakov [KPABG (H) 4385]; **III**: Lower Ponoj River, near Ponoj Village, Schljakov [KPABG (H) 4326]; **IV:** [30]; **V**: [25,27]; **VI**: [12,17,66]; **VIII:** [19,20].

*Cephalozia bicuspidata* (L.) Dumort. s.lat. Common. EU: LC; [1,7]; [82] {***KPABG(H) 9540***}; **I:** [15,65]; **II**: Iokan’ga River basin, Konstantinova, [KPABG(H) 6171]; **III:** Lower Ponoj River, near Ponoj Village, Schljakov [KPABG(H) 3553]; **IV:** [30]; **V**: [24,25,26,27]; **VI**: [12,17,33,35,66]; **VII:** [9,67]; **VIII:** [19,20]; **IX**: [19,20].

*Cephaloziella arctogena* (R.M.Schust.) Konstant. [*Cephaloziella rubella* subsp. *arctogena* (R.M.Schust.) R.M.Schust. & Damsh.]. Rare. EU: VU; **VII:** Konstantinova, Kutsa Sanctuary [KPABG(H) 6129]; **VIII:** [19,20,70]; **IX**: [19,20,70]. {4}

*Cephaloziella divaricata* (Sm.) Schiffn. Frequent. EU: LC; [1]; **I:** [15,65]; **IV**: [18,30]; **V**: [23,24,25,26,27]; **VI**: [12,17,18,33,66]; **VII:** [9,67]; **VIII:** [18,19,20]; **IX**: [19,20]. {4}.

*Cephaloziella elachista* (J.B.Jack ex Gottsche & Rabenh.) Schiffn. Sporadic. EU: VU; MR: 3 (NT); **V**: [23,24,27]; **VI**: [17,83,84]; **VIII:** [19,20,38]; **IX:** [19,20,70]; {4}

*Cephaloziella grimsulana* (J.B.Jack ex Gottsche & Rabenh.) Lacout. Sporadic. EU: DD; [1]; **I:** [15]; **II:** Iokanga River Basin (Sukhaya River Valley), Konstantinova, [KPABG(H) 6177]; **IV:** [30]; **V**: [24,27]; **VI:** [17,66]; **VIII**: [19,20]. {4}

*Cephaloziella hampeana* (Nees) Schiffn. ex Loeske. Sporadic. EU: LC; [1]; **I:** [15]; **II:** Iokanga River Basin (Sukhaya River Valley) Konstantinova [KPABG(H) 6073]; **VI**: [12,17]; **V**: [24,27]; **VIII:** [19]; **IX**: [19]. {4}

*Cephaloziella integerrima* (Lindb.) Warnst. [*Dichiton integerrimus* (Lindb.) H.Buch]. Rare. EU: EN; RF: 3 (VU); MR: 3 (NT); **VI:** [84]; **VII**: [18,46,67,85].

*Cephaloziella polystratosa* (R.M.Schust. & Damsh.) Konstant. [*Cephaloziella divaricata* var. *polystratosa* (R.M.Schust. & Damsh.) Potemkin]. Rare. EU: EN; **VIII**: [19]; **IX:** [19]. {4}.

*Cephaloziella rubella* (Nees) Warnst. Sporadic. EU: LC; [1]; **I:** [1]; **IV**: [30,33]; **V**: Schljakov, near Apatity [KPABG(H) 2160]; [25,26,27]; **VI:** [13,66]; **VIII:** Umba River Basin, Konstantinova [KPABG (H) 121750]; **IX:** Mikkov Isl., Borovichev [INEP153678].

*Cephaloziella spinigera* (Lindb.) Warnst. [*Cephaloziella subdentata* Warnst.]. Sporadic. EU: NT; [1]; **II:** Teriberka Settlement, Borovichev [INEP 152651]; **III:** Rusinga River Valley, Borovichev [INEP152583]; **IV**: [18,30]; **V**: [23,24,25,26,27]; **VI**: [12,13,17,35,66,83]; **VIII:** [19,20]; **IX**: [18,19,20].

*Cephaloziella uncinata* R.M.Schust. Sporadic. EU: NT; **III:** [1,86]; **V**: [24,27,28]; **VIII:** [19,20]; **IX**: [19,20].

*Cephaloziella varians* (Gottsche) Steph. [*Cephaloziella alpina* Douin, *Cephaloziella arctica* Bryhn & Douin, *Cephaloziella grimsulana* (J.B.Jack ex Gottsche & Rabenh.) Lacout. f. arctica (Bryhn & Douin) Schljakov, *Cephaloziella grimsulana* (J.B.Jack ex Gottsche & Rabenh.) Lacout. var. *arctica* var. *nudum* in [1]; *Cephaloziella varians* var. *arctica* (Bryhn & Douin) Damsh., *Cephaloziella varians* var. *scabra* (S.W.Arnell) Damsh.]. Sporadic. EU: LC; [1,7]; [87], {***KPABG K8-2-12***}; **I**: [77]; **III:** Dombrobskaya [KPABG(H) 5476]; **IV**: [30]; **V:** [25,27]; **VI**: [12,33,66]; **VIII:** [19,20]; **IX**: [19,20].

*Chiloscyphus pallescens* (Ehrh.) Dumort. Sporadic. EU: LC; [1]; **I:** [15]; **II:** Sem’ ostrovov Archipelaho, Dombrobskaya [KPABG(H) 4967]; **IV:** [30]; **V:** [25,26,27]; **VI:** [17]; **VIII:** [19,20]; **IX**: [19,20]. *Chiloscyphus pallescens* var. *fragilis* (Roth) Müll.Frib. [*Chiloscyphus fragilis* (Roth) Schiffn.]. Sporadic. EU: LC; [1]; **I:** [15]; **II:** Ramenskaya, Kharlov Isl. [KPABG(H) 4956]; **VI**: [13,17,66]; **VIII**: [19].

*Chiloscyphus polyanthos* (L.) Corda. Frequent. EU: LC; [1,7]; **I:** [15,65]; **II:** Iokanga River Basin, Konstantinova [KPABG(H) 6205]; **III:** Lower Ponoj River, near Ponoj Village, Borovichev [INEP152580]; **IV**: [30]; **V**: [24,25,26,27]; **VI**: [12,13,17,35,66]; **VII:** [9]; **VIII**: [19,20]; **IX:** [19,20].

*Clevea hyalina* (Sommerf.) Lindb. [*Athalamia hyalina* (Sommerf.) S. Hatt.]. Sporadic. EU: LC; MR: 3 (NT); [1,7]; **I**: Lp., [7]; **IV:** [30,72]; **V:** [25,26,27,68]; **VI:** [17,18,33,34,83]; **VII**: [9,67].

*Conocephalum conicum* (L.) Dumort. Sporadic. EU: LC; [1]; **I:** [15,88]; **III:** [88]; **V:** [18,88]; **V**: [24,27]; **VII:** [9,88].

*Conocephalum salebrosum* Szweyk., Buczk. & Odrzyk. Sporadic. EU: LC; **I:** [88]; **V:** [24,26,27,88]; **VII:** [88].

*Crossocalyx hellerianus* (Nees ex Lindenb.) Meyl. [*Anastrophyllum hellerianum* (Nees ex Lindenb.) R.M.Schust.]. Sporadic. EU: LC; MR: 3 (NT); [1]; **IV:** [30,40,46,89]; **V**: [24,26,27]; **VI:** [12,13,17,66,72]; **VII:** [9,46,67,74]; **VIII:** [18,19,20,90]; **IX**: [19,20,41,46].

*Diplophyllum albicans* (L.) Dumort. Sporadic. EU: LC; [1,7]; **I:** Melekhin, Sredniy Peninsula [KPABG(H) 21693]; **IV:** [18,30]; **V:** [5] (**Olenji**); [18,24,25,26,27]; **VI**: [12,17,33,66].

*Diplophyllum obtusifolium* (Hook.) Dumort. Rare. EU: LC; [1]; **V**: [24,25]; **VI**: [1,12,17,66,83].

*Diplophyllum taxifolium* (Wahlenb.) Dumort. Common. EU: LC; [1,7]; **I:** [15]; **II:** Konstantinova, Kildin Isl. [KPABG(H) 4887]; **III:** Rusinga River, Borovichev [INEP152589]; **IV:** [30]; **V:** [24,25,26,27]; **VI**: [12,17,33,66]; **VII:** [9]; **IX**: [19].

*Endogemma caespiticia* (Lindenb.) Konstant., Vilnet & A.V.Troitsky [*Jungermannia caespiticia* Lindenb., *Solenostoma caespiticium* (Lindenb.) Steph.]. Sporadic. EU: LC; [1]; [91] {***KPABG 2/3-02***}; **IV**: [30]; **V:** [25,27]; **VI**: [17,66,83]; **VII**: [92]; **VIII**: [19,20,70]; **IX:** [18,19,20].

*Eremonotus myriocarpus* (Carrington) Lindb. & Kaal. ex Pearson. Sporadic. EU: NT; MR: 3 (NT); **I**: [77]; **V:** [24,27]; **VI:** [93] (as *Sphenolobopsis pearsonii* (Spruce) R.M.Schust.); [33,34,66,94].

*Fossombronia foveolata* Lindb. Rare. EU: LC; MR: 4 (DD); **VI:** [38]; **VIII:** [20].

*Fossombronia incurva* Lindb. Rare. EU: LC; **V**: [27,95].

*Fossombronia wondraczekii* (Corda) Dumort. ex Lindb. Rare. EU: LC; **VI**: [75]. {5}

*Frullania subarctica* Vilnet, Borovich. & Bakalin Rare. EU: DD; **II:** [96] {***KPABG(H) 20639***}.

*Frullania tamarisci* (L.) Dumort. Rare.–EU: LC; MR: 2 (VU); **V:** [96] {***KPABG(H) 18382***}; [23,27].

*Fuscocephaloziopsis affinis* (Lindb. ex Steph.) Váňa & L.Söderstr. [*Cephalozia affinis* Lindb. ex Steph., *Pleurocladula* affinis (Lindb. ex Steph.) Konstant., Vilnet & A.V.Troitsky] Rare. {6}

*Fuscocephaloziopsis albescens* (Hook.) Váňa & L.Söderstr. [*Pleuroclada albescens* (Hook.) Spruce, *Pleurocladula albescens* (Hook.) Grolle]. Common. EU: LC; [1,7]; **I:** [15]; **II:** [7]; **III:** [7]; **IV:** Lavna-Tundra Mts., Konstantinova [KPABG(H) 6094] **V:** [24,25,26,27]; **VI**: [12,17,33,66]; **VIII**: [19,20]; **IX:** Mikkov Isl., Borovichev [INEP153675].

*Fuscocephaloziopsis connivens* (Dicks.) Váňa & L.Söderstr. [*Cephalozia connivens* (Dicks.) Lindb.]. Sporadic. EU: LC; [7,82] {***KPABG(H) 19076***}; **IV:** [97]; **V:** [7,25,27]; **VI**: [12,17,83]; **VIII:** [70].

*Fuscocephaloziopsis leucantha* (Spruce) Váňa & L.Söderstr. [*Cephalozia leucantha* Spruce, *Cephalozia leucantha* var. *robusta* Schljakov]. Common. EU: LC; [1,7]; [82] {***KPABG(H) 8898, 9520, 18054***}; **I:** [15,65]; **II:** Iokanga River Basin (Zolotaya River Basin), Konstantinova, [KPABG(H) 6156]; **III:** Rusinga River, Borovichev [INEP152574]; **V**: [24,25,26,27]; **VI**: [12,17,66,98]; Konstantinova, 2001; **VII:** [9,67] **VIII**: [19,20]; **IX**: [19,20]. {7}

*Fuscocephaloziopsis loitlesbergeri* (Schiffn.) Váňa & L.Söderstr. [*Cephalozia loitlesbergeri* Schiffn., *Pleurocladula loitlesbergeri* (Schiffn.) Konstant., Vilnet & A.V.Troitsky]. Frequent. EU: LC; [1,7]; [82] {***KPABG(H) 11735, 12266***}; **I:** [15]; **II:** Teriberka Settlement, Borovichev [INEP 152652]; **IV:** [30]; **V**: [24,25,27]; **VI:** [12,17]; **VII:** [67]; **VIII**: [19,20]; **IX**: [19,20].

*Fuscocephaloziopsis lunulifolia* (Dumort.) Váňa & L.Söderstr. [*Cephalozia lunulifolia* (Dumort.) Dumort., *Pleurocladula lunulifolia* (Dumort.) Konstant., Vilnet & A.V.Troitsky, *Cephalozia media* Lindb.]. Common. EU: LC; [1,7]; [82] {***KPABG(H) 2853***}; **I:** [15]; **II:** Teriberka Settlement, Borovichev [INEP 152653]; **III:** Rusinga River, Borovichev [INEP152570]; **IV**: [30]; **V**: [24,25,26,27]; **VI:** [12,17,33,66]; **VII:** [9,67]; **VIII:** [19,20]; **IX:** [19,20].

*Fuscocephaloziopsis pleniceps* (Austin) Váňa & L.Söderstr. [*Cephalozia pleniceps* (Austin) Lindb., *Pleurocladula pleniceps* (Austin) Konstant., Vilnet & A.V.Troitsky]. Common. EU: LC; [1,7]; [82] {***KPABG(H) 18350***}; **I:** [15,65]; **II:** Iokanga River Basin (Zolotaya River Basin), Konstantinova, [KPABG(H) 6163]; **III**: [7]; near Ponoj Village, Schljakov [KPABG(H) 3557] **IV**: [30]; **V**: [24,26,27]; **VI**: [12,17,33,66]; **VII:** [9,67]; **VIII**: [19,20]; **IX:** [19,20].

*Geocalyx graveolens* (Schrad.) Nees. Sporadic. EU: LC; **IV:** [18,30]; **V:** [18,24,25,26,27]; **VII:** [18]; **VIII**: [19,20]; **IX**: [19,20,70].

*Gymnocolea inflata* (Huds.) Dumort. Common. EU: LC; [1,7]; [99] {***KPABG(H) 9512***}; **I:** [15,65]; **II:** Teriberka Settlement, Borovichev [INEP 152654]; **III:** Rusinga River, Borovichev [INEP152571]; **IV**: [30]; **V**: [24,25,26,27]; **VI**: [12,17,33,35,66]; **VII:** [9]; **VIII:** [19,20]; **IX**: [19,20].

*Gymnomitrion brevissimum* (Dumort.) Warnst. [*Marsupella brevissima* (Dumort.) Grolle, *Marsupella varians* (Lindb.) Müll. Frib.]. Sporadic. EU: LC; [1,7]; [100] {***KPABG(H) 8171***}; {***KPABG(H) 126739: PQ605318***}; **I: [10,15]; IV**: [18]; **V**: [24,25,26,27]; **VI**: [12,17,33,66,92].

*Gymnomitrion concinnatum* (Lightf.) Corda [*Cesius concinnatus* (Lightf.) Gray]. Common. EU: LC; [1,7]; [100] {***KPABG(H) 8182***}; **I:** [15]; **II:** Murmansk, Konstantinova [KPABG(H) 8087]; **III:** Ponoj valage, Dombrovskaya [KPABG(H) 19171]; **IV**: [30]; **V**: [7,24,25,26,27]; **VI**: [12,17,33,66]; **VII:** [9]; **VIII:** Pestsovye Keivy Mts., Konstantinova [KPABG(H) 6183]; **IX:** Mikkov Isl., Borovichev [INEP153676].

*Gymnomitrion corallioides* Nees [*Cesius corallioides* (Nees) Carruth.]. Frequent. EU: LC; [1,7]; **I:** [77]. **II:** [7]; **III:** Rusinga River, Borovichev [INEP152590]; **IV**: [18,30]; **V**: [18,24,25,26,27]; **VI**, [17,33,66]; **VII:** [9]; **IX:** Mikkov Isl., Borovichev [INEP153677].

*Haplomitrium hookeri* (Lyell ex Sm.) Nees. Sporadic. EU: LC; RF: 3 (VU); MR: 3 (NT); **II:** [37,72]; **IV:** [18,30,46]; **V**: [23,26,27,101]; **VI**: [66]; **VIII**: [19,20,70,90]; **IX:** Kovdozero Lake Basin, Tolvand Valley, Konstantinova [KPABG(H) 14140].

*Harpanthus flotovianus* (Nees) Nees [*Harpanthus flotovianus* var. *chiloscyphoides* Arnell, *Harpanthus flotovianus* var. *latifolius* Jørg., *Harpanthus flotovianus* var. *retusus* Jørg.]. Common. EU: NT; [1,7]; **I:** [15]; **II:** Teriberka Settlement, Borovichev [INEP 152656]; **IV**: [30]; **V:** [24,25,26,27]; **VI**: [12,17,33,35,66]; **VII:** [9]; **VIII**: [19,20]; **IX:** at the base of Gremyashka Mt., valley of Kanda River, Konstantinova [KPABG(H) 6133].

*Heterogemma laxa* (Lindb.) Konstant. & Vilnet [*Lophozia laxa* (Lindb.) Grolle, *Lophozia marchica* (Nees ex Limpr.) Steph., *Massularia laxa* (Lindb.) Schljakov, *Schistochilopsis laxa* (Lindb.) Konstant.]. Sporadic. EU: VU; MR: 3 (NT); [100] {***KPABG(H) 2694***}; **V**: [1,24,26,27]; **VI**: [72,84]; **VII:** [18,67,102]; **VIII:** [18,19,20,38,70,72]; **IX:** [18].

*Hygrobiella laxifolia* (Hook.) Spruce. Sporadic. EU: LC; [1,7]; [103] {***KPABG(H) 5171, 5187, 6103, 6984, 8112, 9487, 12761, 12766, 19152, 20180, 123339***}; **I:** [15,77]; **II:** Iokan’ga River Basin, Konstantinova [KPABG(H) 6177]; **IV:** [18]; **V**: [24,25,27]; **VI**: [12,17,33,66]; **IX:** Kanda River Basin, Konstantinova [KPABG(H) 5177].

*Isopaches alboviridis* (R.M.Schust.) Schljakov. Rare. EU: DD; **III**: [86].

*Isopaches bicrenatus* (Schmidel ex Hoffm.) H.Buch [*Lophozia bicrenata* (Schmidel ex Hoffm.) Dumort.]. Frequent. EU: LC; [1]; [] [***KPABG(H) 124358***]; **I:** [77]; **II:** Teriberka Settlement, Borovichev [INEP 152662]; **III:** Rusinga River, Borovichev [INEP152586]; **IV**: [30]; **V**: [24,25,26,27]; **VI**: [17,66]; **VII:** near Alakurtti settlement, Schljakov [KPABG(H) 3901]; **VIII**: [19,20]; **IX**: at the base of Gremyashka Mt., valley of Kanda River, Konstantinova [KPABG(H) 3891].

*Isopaches decolorans* (Limpr.) H.Buch [*Lophozia decolorans* (Limpr.) Steph.]. Rare. EU: DD; RF: 3 (VU); MR: 3 (NT); [1]; **V**: [28,72,73]; **VI**: [12,66,92].

*Jungermannia atrovirens* Dumort. [*Jungermannia lanceolata* var. *atrovirens* (Dumort.) Damsh.]. Sporadic. EU: LC; **III:** valley of Ponoj River, Schljakov [KPABG(H) 4107]; **VI:** Khibiny Mts., Schljakov [KPABG(H) 4117]; **VII:** [9,67]; **VIII**: [19] (as cf.). {8}

*Jungermannia borealis* Damsh. & Váňa [Aplozia oblongifolia auct. non. Müll.Frib. Jungermannia oblongifolia sensu auct., Solenostoma oblongifolium sensu auct., Jungermannia karl-muelleri sensu Grolle.] Sporadic. EU: LC; [1]; [104] {***KPABG(H) 595***}. **I:** [15]; **III:** Lower Ponoj River, near Ponoj Village, Schljakov [KPABG(H) 4105]; **IV**: [18,30]; **V**: [24,25,26,27]; **VI:** [12,17,66]; **VII**: Schljakov, Ontonjoki River Basin [KPABG(H) 4124]; **VIII**: [19,20].

*Jungermannia eucordifolia* Schljakov [Aplozia cordifolia Dumort., Jungermannia exsertifolia subsp. cordifolia (Dumort.) Váňa, Jungermannia cordifolia Hook., Solenostoma cordifolium (Dumort.) Steph.]. Frequent. EU: LC; [1]; [104] {***KPABG(H) 9596***}; **I:** [15]; **III:** Lower Ponoj River, Orlov cape, Schljakov [KPABG(H) 4139]; **IV**: [18,97]; **V**: [24,25,26,27]; **VI**: [12,17,33]; **VII:** [9].

*Jungermannia polaris* Lindb. [*Aplozia polaris* (Lindb.) Bryhn et Kaal., *Jungermannia pumila* subsp. *polaris* (Lindb.) R.M.Schust., *Solenostoma schiffneri* (Loitl.) Müll.Frib.]. Sporadic. EU: LC; [1]; [104] {***KPABG(H) 8170, 6939***}; **I**: [77]; **III:** Lower Ponoj River, Orlov cape, Schljakov [KPABG(H) 4148]; **IV:** Lavna Tundra Mts., Konstantinova {KPABG(H) 4139] **V**: [24,25,27]; **VI:** [12,17,33,66]. {9}

*Jungermannia pumila* With. [*Aplozia pumila* (With.) Dumort., *Jungermannia pumila* var. *alpestris* Gottsche & Rabenh., *Solenostoma pumilum* (With.) Müll.Frib.]. Sporadic. EU: LC; [1,7]; **I:** [15]; **II:** Kildin Isl., Konstantinova [KPABG(H) 5008]; III: [7]; **IV**: [18,30]; **V**: [24,27]; **VI**: [12,17,33,66]; **VII:** [9]; **VIII**: [19]; **IX:** Kovdozero Lake Basin, Konstantinova [KPABG(H) 14127]. {9}.

*Kurzia pauciflora* (Dicks.) Grolle. Sporadic. EU: LC; MR: 3 (NT); **II:** [32,72,81]; **IV:** [40,72,89]; **V:** [23,24,27]; **VI:** [43,81]; **VIII:** Schljakov, Ponoj River Basin, near Krasnoshshel’e [KPABG(H) 5108]; [38]; **IX**: [19,20].

*Lejeunea cavifolia* (Ehrh.) Lindb. Rare. EU: LC; MR: 2 (VU); **V:** [23,25,27,81].

*Lepidozia reptans* (L.) Dumort. Sporadic. EU: LC; [1]; **I:** [15]; **II:** Schljakov, Krasnoshshel’e [KPABG(H) 5102]; **III:** Lower Ponoj River, near Ponoj Village, Borovichev [INEP152581]; **IV:** [30]; **V:** [24,25,26,27]; **VI:** [12,17]; **VII:** [9,67]; **VIII**: [19,20]; **IX:** [19,20].

*Liochlaena lanceolata* Nees [*Aplozia lanceolata* (Nees) Müll.Frib., *Jungermannia lanceolata* sensu Schrad., *Jungermannia lanceolata* auct. non L., *Jungermannia leiantha* Grolle, *Jungermannia subulata* var. *leiantha* (Grolle) Damsh.]. Sporadic. EU: LC; [1]; [100] **{*KPABG(H) 9550*}**, **{*KPABG(H) 12258, 16255 GenBank (https://www.ncbi.nlm.nih.gov/*)}**; **I:** [15]; **IV:** [18,30]; **V**: [18,24,25,26,27]; **VI**: [17]; **VII:** Konstantinova, Kutsa Sanctuary [KPABG(H) 6121]; **VIII**: [19,20]; **IX:** [19,20].

*Lophocolea heterophylla* (Schrad.) Dumort. [*Chiloscyphus profundus* (Nees) J.J.Engel & R.M.Schust.]. Sporadic. EU: LC; [1]; **I:** [15]; **III:** Lower Ponoj River, near Ponoj Village, Borovichev [INEP152584]; **IV**: [18,30]; **V**: [23,24,25,26,27]; **VI:** [13,17]; **VII:** Konstantinova, Kutsa Sanctuary [KPABG(H) 6120]; **VIII:** [19,20]; **IX:** [19,20].

*Lophocolea minor* Nees [*Chiloscyphus minor* (Nees) J.J.Engel & R.M.Schust.]. Sporadic. EU: LC; [1,7]; **I:** [15]; **II:** Schljakov, Krasnoshshel’e [KPABG(H) 4955]; **IV**: [18,30]; **V**: [23,24,25,27]; **VI**: [12,13,17]; **VII:** [9]; Konstantinova, Kutsa Sanctuary [KPABG(H) 6121]; **VIII**: [19,20]; **IX:** Kanda River Valley, at base of Gremyashka Mt., Konstantinova [KPABG(H) 4937].

*Lophozia ascendens* (Warnst.) R.M.Schust. [*Lophozia gracillima* H.Buch]. Sporadic. EU: LC; MR: 3 (NT); **IV**: [30,72]; **V**: [23,24,26,27]; **VII**: [1,9,22,46]; **VIII**: [22]; **IX:** [22].

*Lophozia guttulata* (Lindb. & Arnell) A.Evans [*Lophozia porphyroleuca* (Nees) Schiffn., *Lophozia longiflora* auct. (sensu [105,106,107]), *Lophozia longiflora* var. *guttulata* (Lindb. & Arnell) Schljakov]. Frequent. EU: LC; **II:** near Murmansk City, A.V. Dombrovskaya [KPABG(H) 19169]; **IV**: [30]; **V**: [22,24,25,26,27]; **VI**: [22]; **VII:** [9]; **VIII**: [22]. {10}

*Lophozia longiflora* (Nees) Schiffn. [*Lophozia ventricosa* var. *longiflora* (Nees) Macoun, *Lophozia ventricosa* var. *uliginosa* auct. (sensu [105,106,107])]. Frequent. EU: LC; [1]; **I:** [15,22]; **II**: [22]; **III:** [22]; **IV:** [30]; **V**: [24,25,26,27]; **VI**: [12,17,22,33]; **VIII:** [19,20]; **IX**: [19,20]. {10}

*Lophozia murmanica* Kaal. [*Lophozia wenzelii* var. *groenlandica* (Nees) Bakalin (sensu [108]), *Lophozia confertifolia* auct. (sensu [109]), *Lophozia heteromorpha* R.M.Schust. & Damsh.]. Frequent. EU: DD; **I:** [15,22]; **II**: Konstantinova, Iokan’ga River Basin [KPABG(H) 6173]; **III:** Ponoj Village, Schljakov [KPABG(H) 3743 **IV**: [30]; **V**: [24,25,26,27]; **VI:** [12,17,22,33,35,66]; **VII:** Kutsa Sanctuary, Schljakov [KPABG(H) 3510]; **VIII**: [19,20]; **IX**: [19,20,22]. {10}

*Lophozia savicziae* Schljakov [*Lophozia silvicola* var. *grandiretis* H.Buch & S.W.Arnell, *Lophozia ventricosa* var. *grandiretis* (H.Buch & S.W.Arnell) R.M.Schust. & Damsh., *Lophozia murmanica* auct. (sensu [110])]. Sporadic. EU: VU; [1]; **I:** [65]; **II**: [22]; **III:** [1]; Lower Ponoj near Ponoj Village, Schljakov [KPABG(H) 3742]; **IV:** [18,30]; **V:** [24,26,27]; **VI**: [111]: **type description** [Schljakov [KPABG(H) 3752]; [12,17,22,33,66]; **VII:** Schljakov, valley of Tumcha River [KPABG(H) 9354]; **VIII:** Pestsovye Keivy Mts., Konstantinova, [KPABG(H) 6185].

*Lophozia schusteriana* Schljakov [*Lophozia groenlandica* sensu [112]]. Rare. EU: LC; [100] {***KPABG(H) 9331***}; **V**: [22,28]; **VI**: [22,66]; **VIII:** Kashkarantsy, Schljakov [KPABG(H) 18340].

*Lophozia silvicola* H.Buch [*Lophozia ventricosa* var. *silvicola* (H.Buch) E.W.Jones]. Sporadic. EU: LC; **I**: [22,77]; **II**: [22]; **IV**: [30]; **V**: [22,24,25,26,27]; **VI:** [22,66]; **VII:** [9]; **VIII:** Konstantinova, Kamennik Mt. [KPABG(H) 18081]; **IX:** Mikkov Isl., Borovichev [INEP153680].

*Lophozia silvicoloides* N.Kitag. Rare. EU: DD; [100] {***KPABG(H) 8088***}; **II**: [22].

*Lophozia ventricosa* (Dicks.) Dumort. [*Lophozia groenlandica* auct. (sensu [113,114]), *Lophozia confertifolia* auct. (sensu [114,115,116]), *Lophozia murmanica* auct. (sensu [117]), *Lophozia ventricosa* var. *confusa* R.M.Schust.] All provinces. Since, at different times, the interpretations of the species in the territory of the region differed, here we refrain from specifying the distribution of the species in the floristic regions. EU: LC. {10}

*Lophozia wenzelii* (Nees) Steph. [*Lophozia groenlandica* auct. (sensu [108,116,118]), *Lophozia confertifolia* Schiffn. (sensu [108,119]), *Lophozia ventricosa* var. *uliginosa* Breidl. ex Schiffn., *Lophozia iremelensis* Schljakov]. Common. EU: LC; [1]; [100] {***KPABG(H) 9329, 9697***}; **I:** [15]; **II:** [22]; **IV**: [22]; **V**: [24,25,27]; **VI**: [12,17,66]; **VII:** [9]; **VIII**: [19,20,22]; **IX:** Mikkov Isl., Borovichev [INEP153681].

*Lophoziopsis excisa* (Dicks.) Konstant. & Vilnet [*Lophozia excisa* (Dicks.) Dumort. *Lophozia excisa* var. *cylindracea* (Dumort.) Müll.Frib.]. Sporadic. EU: LC; [1]; [100] {***KPABG(H) 6146***}; **I:** [77]; **II:** [22]; **III:** [22]; **IV**: [30]; **V**: [25,26,27]; **VI**: [12,17,33,66]; **VII:** Kutsa Sanctuary, Schljakov [KPABG(H) 3714]. **VIII**: [19,20]; **IX**: [52] [KPABG(H) 123449].

*Lophoziopsis excisa* var. *elegans* (R.M.Schust.) Konstant. & Vilnet. Rare. EU: NE; [120] {***KPABG(H) 124364***}; **V**: [25,27,28]; **VI:** [120].

*Lophoziopsis jurensis* (Meyl. ex Müll.Frib.) Mamontov & Vilnet [*Lophozia jurensis* Meyl. ex Müll. Frib., *Lophozia propagulifera* auct. eur., *Lophoziopsis propagulifera* auct. eur., *Lophozia latifolia* R.M.Schust., *Lophoziopsis latifolia* (R.M.Schust.) Köckinger]. Sporadic. EU: LC; [1]; **II:** [92]: **III**: [92]: Lower Ponoj River, Dombrovskaja [KPABG(H) 18334]; [22]; **V**: [22,25,27,28]; **VI:** [12,17,22,66,92].

*Lophoziopsis longidens* (Lindb.) Konstant. & Vilnet [*Lophozia longidens* (Lindb.) Macoun]. Frequent. EU: LC; [1]; [100] {***KPABG(H) 8110***}; [121] {***KPABG(H) 125223***}; **I:** [15]; **II:** [22]; **III:** [22]; **IV:** [22,30]; **V**: [22,24,25,26,27]; **VI**: [12,17,22,66]; **VII:** [9]; **VIII**: [19,20]; **IX**: Kanda River Valley Konstantinova, at base of Gremyashka Mt.[KPABG(H) 5575].

*Lophoziopsis pellucida* (R.M. Schust.) Konstant. & Vilnet. [*Lophozia pellucida* R.M. Schust.]. Rare. EU: VU; [122] {***KPABG(H) 11721***}; **V:** [81].

*Lophoziopsis polaris* (R.M.Schust.) Konstant. & Vilnet [*Lophozia polaris* (R.M.Schust.) R.M.Schust. & Damsh., *Lophozia major* (C.E.O.Jensen) Schljakov, *Lophozia uncinata* Schljakov]. Rare. EU: VU; [1]; **V:** [22]; **VI**: [17,22]; **VII:** [111].

*Lophoziopsis rubrigemma* (R.M. Schust.) Konstant. & Vilnet. [*Lophozia rubrigemma* R.M. Schust., *Lophozia pellucida* R.M. Schust. var. *rubrigemma* (R.M.Schust.) Bakalin] Rare. EU: DD; **I**: [77]; **II**: [22]; **VIII:** [70].

*Mannia gracilis* (F.Weber) D.B.Schill & D.G.Long [*Asterella ludwigii* (Schwägr.) Underw., *Asterella gracilis* (F.Weber) Underw.] Sporadic. EU: LC; [7]; [123] {***KPABG(H) 16249, 18369***, ***P-305-6-13***}; **II:** [124]; F.Nylander [KPABG(H): 20662]; **V**: [7] (Lim: as *Asterella ludwigii*); [24,26,27,124]; **IV**: [30]; **VIII:** [124].

*Mannia pilosa* (Hornem.) Frye & L.Clark [*Neesiella pilosa* (Hornem.) Schiffn.]. Rare. EU: LC; MR: 3 (NT); **IV**: [30,72]; **V**: [25,27]; **VII**: [9,67].

*Mannia triandra* (Scop.) Grolle. Rare. EU: VU; RF: 3 (VU); [125]; **V**: [24,25,124].

*Marchantia polymorpha* L. s lat. EU: LC; [1,7]; **VI**: [17]; **VII:** [9,67]. {12}

*Marchantia polymorpha* subsp. *polymorpha* [*Marchantia aquatica* (Nees) Burgeff]. Rare. EU: NE; **V:** [24,27]; **VI:** [1].

*Marchantia polymorpha* subsp. *montivagans* Bischl. & Boissel.- Dub. [*Marchantia alpestris* (Nees) Burgeff]. Common.EU: NE; [1]; **I:** [15,65]; **II:** Teriberka, M.Kh.Kachurin, M.L.Kachurina [KPABG(H) 5668]; **IV:** [97]; **V**: [24,25,26,27]; **VI:** [12,17,66]; **VII:** Tumcha River, Dombrovskaya [KPABG(H) 5696]; **VIII:** Schljakov [KPABG(H) 5699].

*Marchantia polymorpha* subsp. *ruderalis* Bischl. et Boissel.-Dub. [*Marchantia latifolia* Gray]. Common. EU: LC; [1,7]; **I:** [15]; **II**: Kildin, Konstantinova [KPABG(H) 5008]; **III:** Lower Ponoj River, near Ponoj Village, Borovichev [INEP152585]; **IV:** [30]; **V**: [25,26,27]. **VI:** [12,17,35,66]; **VII**: Kutsa Sanctuary, Schljakov [KPABG(H) 5714]; **VIII**: [19,20]; **IX**: [19,20].

*Marchantia quadrata* Scop. [*Preissia quadrata* (Scop.) Nees]. Sporadic. EU: LC; [1,7]; [126] {***KPABG(H) 18347***}; {***KPABG(H) 6932, 18347, 12754 GenBank https://www.ncbi.nlm.nih.gov/)***}; **I:** [15]; **II:** Kildin, Konstantinova [KPABG(H) 5639] **III:** Rusinga, Schljakov [KPABG(H) 3449]; **IV**: [30]; **V**: [24,25,26,27]; **VI**: [12,17,33]; **VII:** [9]; **VIII:** [19].

*Marsupella apiculata* Schiffn. [*Gymnomitrion apiculatum* (Schiffn.) Müll.Frib.] [1,7]; EU: LC; **I**: [7,77]; **II**: Teriberka River Basin, Schljakov [KPABG(H) 3783]; **IV:** [18]; **V:** [7,24,25,26,27]; **VI:** [12,17,33,66]; **VII:** Kutsa Sanctuary, Pukhyakuru, Schljakov [KPABG(H) 6051]; **VIII:** Pana Tundra, Kamennik Mt., Konstantinova [KPABG(H) 18037].

*Marsupella aquatica* (Lindenb.) Schiffn. [*Marsupella robusta* (De Not.) A.Evans *Marsupella robusta* (De Not.) A.Evans f. pearsonii (Schiffn.) Schljakov Evans *Marsupella emarginata* subsp. *aquatica* (Lindenb.) Meyl., *Marsupella emarginata* var. *aquatica* (Lindenb.) Dumort.]. Sporadic. EU: LC; [1]; [100] {***KPABG(H) 6090***}; **I:** [15]; **II:** Iokan’ga River Basin, Konstantinova [KPABG(H) 6202]; **IV**: [18,97]; **V**: [24,25,26,27]; **VI:** [12,17]; **VII**: [18].

*Marsupella boeckii* (Austin) Lindb. ex Kaal. Sporadic. EU: LC; [1,7]; [100] {***KPABG(H) 8184***}; **I**: [77]; **III:** Lumbovka Bay, Borovichev [KPABG(H) 18653]; **IV**: [18,30]; **V**: [24,25,26,27]; **VI**: [17,33,66].

*Marsupella condensata* (Ångstr. ex C.Hartm.) Lindb. ex Kaal. Sporadic. EU: VU; [1,7]; **IV**: [18]; **V**: [24,25,27]; **VI**: [12,17,33,66]; **VIII:** Pestsovye Keivy Mts., Konstantinova [KPABG(H) 6184].

*Marsupella emarginata* (Ehrh.) Dumort. [*Marsupella emarginata* var. *pearsonii* (Schiffn. ex Macvicar) Jørg.]. Sporadic. EU: LC; [1,7]; [100] {***KPABG(H) 8070***}; **III:** Lumbovka Bay, Borovichev [KPABG(H) 120323]; **IV**: [30]; **V**: [24,25,27]; **VI**: [17,33,66]; **VII:** [9,67]; **VIII**: Pana Tundra, Kamennik Mt. Konstantinova [KPABG(H) 18069]. {13}

*Marsupella sparsifolia* (Lindb.) Dumort. Rare. EU: NT; **II**: [1]; **IV**: [18]; **VI**: [66].

*Marsupella sphacelata* (Giesecke ex Lindenb.) Dumort. Sporadic. EU: LC; [1,7]; [100] {***KPABG(H) 11225***}; **I:** [15]; **II:** Kildin, Konstantinova [KPABG(H) 4682]; **IV:** Chiltald, Konstantinova [KPABG(H) 4203]; **V:** [24,25,27]; **VI:** [12,13,66]; **VIII**: Kashkarantsy, Schuster, Konstantinova [KPABG(H) 6030].

*Marsupella sprucei* (Limpr.) Bernet [*Marsupella ustulata* Spruce, *Marsupella sprucei* var. *neglecta* (Limpr.) Damsh., *Marsupella sprucei* var. *ustulata* (Spruce) Damsh.]. Sporadic. EU: LC; [1]; **I**: [77]; **IV:** [18]; **V**: [24,25,27]; **VI**: [17,66]; **VIII:** Kamennik Mt., Konstantinova [KPABG(H) 18037].

*Mesoptychia badensis* (Gottsche ex Rabenh.) L.Söderstr. & Váňa [*Lophozia badensis* (Gottsche ex Rabenh.) Schiffn., *Leiocolea badensis* (Gottsche ex Rabenh.) Jørg., *Lophozia badensis* var. *obtusiloba* (Bernet) Schiffn.]. Sporadic. EU: LC; MR: 3 (NT); [1,7]; **II:** [7]; **III:** [36,72,73]; **IV**: [30,72,89]; **V**: [25,27]; **VI:** [33], 2020; **VII:** [9,67].

*Mesoptychia bantriensis* (Hook.) L.Söderstr. & Váňa [*Lophozia bantriensis* (Hook.) Steph., *Leiocolea bantriensis* (Hook.) Jørg.]. Sporadic. EU: LC; [1]; **I:** [15]; **IV**: [18,30]; **VI**: [13,18,66]; **VII:** [9,67]; **VIII:** [19,20].

*Mesoptychia collaris* (Nees) L.Söderstr. & Váňa [*Leiocolea collaris* (Nees) Schljakov, *Lophozia alpestris* (Schleich. ex F.Weber) A.Evans nom. rej., *Leiocolea alpestris* (Schleich. ex F.Weber) Isov., *Lophozia alpestris* var. *libertiae* (Huebener) Damsh., *Leiocolea muelleri* (Nees) Jørg.]. Sporadic. EU: LC; [1,7]; **I:** [15]; **III:** [7]; Ponoj, Schljakov [KPABG(H) 3625]; **IV:** [7]; **V:** Umba River Basin, Konstantinova, [KPABG(H) 121732]; **VI**: [13,17]; **VII:** [9,67]; **VIII:** [19,20].

*Mesoptychia gillmanii* (Austin) L.Söderstr. & Váňa [*Lophozia gillmanii* (Austin) R.M.Schust., *Leiocolea gillmanii* (Austin) A.Evans, *Lophozia gillmanii* var. *acutifolia* (Limpr.) R.M.Schust., *Lophozia kaurinii* (Limpr.) Steph.]. Sporadic. EU: VU; [1]; **I:** [15]; **II:** Teriberka Settlement, Borovichev [INEP 152666]; **III:** Lower Ponoj River Basin, Schljakov [KPABG(H) 3625]; **IV:** [30]; **V:** [24,25,26,27]; **VI:** [12,33,66]; **VII:** [9,67]; **VIII**: [19,20].

*Mesoptychia heterocolpos* (Thed. ex Hartm.) L.Söderstr. & Váňa [*Lophozia heterocolpos* (Thed. ex Hartm.) M.Howe, *Leiocolea heterocolpos* (Thed. ex Hartm.) H.Buch]. Frequent. EU: LC; [7]; [1]; **I:** [15]; **II:** Kildin Isl. Schljakov [KPABG(H) 3409]; **III:** [7]; Lower Ponoj River Basin, Schljakov [KPABG(H) 5640]; **IV:** [30]; **V**: [24,25,26,27]; **VI:** [12,17,33,66]; **VII:** [9,67]; **VIII:** [19,20]; **IX:** Kovdozero Lake Basin, Konstantinova [KPABG(H) 14118].

*Mesoptychia heterocolpos* var. *arctica* (S.W.Arnell) L.Söderstr. & Váňa [*Lophozia heterocolpos* var. *arctica* (S.W.Arnell) R.M.Schust. & Damsh., *Leiocolea heterocolpos* var. *savicziae* Schljakov]. Rare. EU: NE; **I:** Srednij Peninsula, Konstantinova [KPABG(H)126018]; **VII:** Kutsa Sanctuary, Nivajarvi, Konstantinova [KPABG(H) 6112]; **VIII**: [19]; **IX:** Kanda River Basin, Konstantinova [KPABG(H) 5054]. {14}

*Mesoptychia heterocolpos* var. *harpanthoides* (Bryhn & Kaal.) L.Söderstr. & Váňa [*Lophozia heterocolpos* var. *harpanthoides* (Bryhn & Kaal.) R.M.Schust.] III: [1]. {14}

*Mesoptychia rutheana* (Limpr.) L.Söderstr. & Váňa [*Lophozia rutheana* (Limpr.) M.Howe, *Leiocolea rutheana* (Limpr.) Müll.Frib., *Lophozia schultzii* Schiffn.]. Sporadic. EU: LC; [1]; [127] {***KPABG(H) 2684***}; **III**: [9]: Orlov Cape, Gogulikha; [7]; **IV:** [18,30]; **V:** [25,27]; **VI**: [17,35,66]; **VII:** [9,67]; **VIII:** [19,20].

*Metzgeria furcata* (L.) Corda. Sporadic. EU: LC; MR: 3 (NT); [1,7]; **I:** [7]; **II:** [37,72]; **III:** [36,72]; **IV**: [18,30,72,89]; **V:** [24,25,27,31,72]; **VI:** [33,34]; **VII:** [9,46,67,74]; **VIII**: [19,20,70]; **IX:** [42].

*Moerckia flotoviana* (Nees) Schiffn. [*Moerckia hibernica* (Hook.) Gottsche auct.]. Sporadic. EU: LC; [1]; [128] {***KPABG(H) 20670, 20774***}; **IV**: [18]; **V**: [18,24,25,26,27]; **VI:** [12,33,35]; **VII:** [9,67]; **VIII:** [19,20,70]. {15}

*Mylia anomala* (Hook.) Gray [*Leptoscyphus anomalus* (Hook.) Lindb.]. Frequent. EU: LC; [1,7]; **I:** [15]; **II:** Iokan’ga River Basin, Konstantinova [KPABG(H) 6171]; **III:** Lower Ponoj Schljakov[KPABG(H) 5353]; **IV**, [30]; **V:** [24,25,26,27]; **VI**: [12,17,35,66]; **VII:** [9,67]; **VIII**: [19,20]; **IX**: Valley of Kovdozero Lake, Ramenskaya [KPABG(H) 4414].

*Mylia taylorii* (Hook.) Gray. Sporadic. EU: LC; [1,7]; **I:** [15]; **II:** [7]; Teriberka River Basin, Eriveiv Mt., Schljakov [KPABG (H) 4419]; **IV:** [18,30]; **V**: [18,24,25,27]; **VI:** [18]; **VIII:** Luven’ga tundra Mts., Konstantinova [KPABG(H) 4421]; **IX:** Mikkov Isl., Borovichev [INEP153683].

*Nardia breidleri* (Limpr.) Lindb. Sporadic. EU: LC; [1,7]; 5; **I:** [15,46]; **IV**: [18,30,72,89]; **V:** [24,27]; **VI:** [12,17,33,34,66].

*Nardia geoscyphus* (De Not.) Lindb. Common. EU: LC; [1,7]; **I:** [15]; **II:** Kildin Isl., Konstantinova [KPABG(H) 4315]; **III:** Lower Ponoj River, Schljakov [KPABG(H) 4531 **IV:** [30]; **V:** [24,25,27,28]; **VI**: [12,17,33,66]; **VII:** [9]; [67]; **VIII**: [19,20]; **IX**: Kovdozero Lake Basin, Konstantinova [KPABG(H) 14127].

*Nardia geoscyphus* var. *bifida* R.M. Schust. Rare. EU: NE; [1]; **VI:** [66,129].

*Nardia insecta* Lindb. [*Nardia geoscyphus* f. *insecta* (Lindb.) Arnell.]. Sporadic. EU: LC; [1]; **I:** Near Linakhamari, Schljakov [KPABG(H) 4376]; **V**: [24,26,27]; **VI:** [84]; **VII:** Near Alakurtti Settlement, Schljakov [KPABG(H) 3901]; **VIII:** near Kashkarantsy Village, Schljakov [KPABG(H) 6026].

*Nardia japonica* Steph. Sporadic. EU: LC; **IV**: [30,85]; **V**: [26,27]; **VI:** [18,70,84]. {16}.

*Nardia scalaris* Gray. Sporadic. EU: LC; [1,7]; **I:** [15]; **II:** Middle Teriberka River Schljakov [KPABG(H) 4385]; **III:** Lower Ponoj River, Schljakov [KPABG(H) 4386]; **V**: [24,25,26,27]; **VI:** [12,17,66]; **VII:** [67]; Tumcha River Basin, Schljakov [KPABG(H) 4397].

*Neoorthocaulis attenuatus* (Mart.) L.Söderstr., De Roo & Hedd. [*Barbilophozia attenuata* (Mart.) Loeske, *Orthocaulis attenuatus* (Mart.) A.Evans, *O. gracilis* (Lindenb.) H.Buch,, *Lophozia attenuata* (Mart.) Dumort. *Lophozia gracilis* (Lindenb.) Steph.]. Frequent. EU: LC; [1,7]; [100] {***KPABG(H) 6174***}; [121] {***KPABG(H) 2325***]; [***KPABG(H) 9504: PQ585838, PQ605317***}; **I:** [7]; **II:** Sukhaya River, Iokan’ga River Basin, Konstantinova [KPABG(H) 6174]; **III:** Lumbovka Bay, Borovichev [KPABG(H)18647]; **IV**: [18,30]; **V**: [25,27]; **VI:** [13,17]; **VII:** [9,67]; **VIII**: [19,20]; **IX**: [19,20].

*Neoorthocaulis binsteadii* (Kaal.) L.Söderstr., De Roo & Hedd. [*Barbilophozia binsteadii* (Kaal.) Loeske, *Orthocaulis binsteadii* (Kaal.) H.Buch, *Lophozia binsteadii* (Kaal.) A.Evans]. Frequent. EU: LC; [1,7]; [100] {***KPABG(H)9676***}; [121] {***KPABG(H)481***}; **I:** [15]; **II:** Iokan’ga River Basin, Schljakov [KPABG(H) 2333]; **III:** Rusinga River, Borovichev [INEP 152572]; **IV**: [30]; **V:** [24,25,26,27]; **VI:** [12,13,17,66]; **VII:** [9,67]; **VIII:** [19,20]; **IX**: [19,20].

*Neoorthocaulis floerkei* (F.Weber & D.Mohr) L.Söderstr., De Roo & Hedd. [*Barbilophozia floerkei* (F.Weber & D.Mohr) Loeske, *Orthocaulis floerkei* (F.Weber & D.Mohr) H.Buch, *Lophozia floerkei* (F.Weber & D.Mohr) Schiffn.]. Frequent. EU: LC; [1,7]; [100] {***KPABG(H)9510***}; [121] {***KPABG(H) 9566***}; **I:** [15]; **II:** Lower Teriberka River, Mamontov [INEP 400395]; **IV**: [30]; **V:** [24,25,26,27]; **VI**: [12,17,66]; **VII:** [9,67]; **VIII:** Konstantinova, 1998; **IX**: [19,20].

*Nowellia curvifolia* (Dicks.) Mitt. Rare. EU: LC; **II:** [73]; **V:** [23].

*Obtusifolium obtusum* (Lindb.) S.W.Arnell [*Lophozia obtusa* (Lindb.) A.Evans]. Sporadic. EU: LC; [1,7]; [127] {***KPABG(H)8775***}; **I:** [15,65]; **II:** Teriberka Settlement, Borovichev [INEP 152674]; **IV**: [30]; **V**: [24,25,26,27]; **VI:** [12,17,66]; **VII:** [9,67]; **VIII**: [19,20]; **IX**: Mikkov Isl., Borovichev [INEP153684]. {17}

*Odontoschisma elongatum* (Lindb.) A. Evans. Common. EU: LC; [1,7]; [82] {***KPABG(H)18025***}; **I:** [15]; **II:** [7]; Teriberka River Basun, Konstantinova [KPABG(H)1867]; **III:** Rusinga River, Borovichev [INEP152567]; **IV**: [30]; **V**: [24,25,26,27]; **VI**: [12,17,33,66]; **VII:** Kutsa Sanctuary, Konstantinova [KPABG(H) 5464]; **VIII:** [19,20]; **IX:** [19,20].

*Odontoschisma fluitans* (Nees) L.Söderstr. & Váňa. [*Cladopodiella fluitans* (Nees.) Jørg., *Cephalozia fluitans* (Nees) Spruce]. Frequent. EU: LC; [1,7]; [82] {***KPABG(H)124336***}; **I:** [15]; **II:** Valley of Teriberka River, Mamontov [KPABG(H) 124336]; **III:** Rusinga River, Borovichev [INEP152573]; **IV**: [18,30]; **V:** [24,25,26,27]; **VI**: [13,17]; VII: Kutsa Sanctuary, Konstantinova [KPABG(H) 1939]; **VIII**: [19,20]; **IX**: [19,20].

*Odontoschisma francisci* (Hook.) L.Söderstr. & Váňa [*Cladopodiella francisci* (Hook.) Jørg., *Cephalozia francisci* (Hook.) Dumort.]. Sporadic. EU: NT; [1,7]; [82] {***KPABG(H)6094, 6185***}; **II:** Iokan’ga River Basin, Konstantinova [KPABG(H) 6164]; **IV**: [18,30]; **V**: [24,25,27]; **VI:** [12,13,17]; **VII:** Kutsa Sanctuary, Konstantinova [KPABG(H) 6123]. **VIII:** Panskie Tundra Mts., Kamennik Mt., Konstantinova [KPABG(H) 18037].

*Odontoschisma macounii* (Austin) Underw. Sporadic. EU: LC; [1,7]; [82] {***KPABG(H)6445***}; **I**: [77]; **II:** Teriberka Settlement, Borovichev [INEP 152664]; **III:** Rusinga River, Borovichev [INEP152560]; **IV**: [30]; **V**: [23,24,25,26,27]; **VI**: [12,17,33]; **VII:** [9,67]; **VIII:** Panskie Tundra, Kamennik Mt., Konstantinova [KPABG(H) 18025].

*Oleolophozia perssonii* (H.Buch & S.W.Arnell) L.Söderstr., De Roo & Hedd. [*Lophozia perssonii* H.Buch & S.W.Arnell, *Lophoziopsis perssonii* (H.Buch & S.W.Arnell) Konstant. & Vilnet]. Rare. EU: LC; MR: 2 (VU); **IV:** [30,72]; **V**: [27,28,75].

*Orthocaulis atlanticus* (Kaal.) H.Buch [*Barbilophozia atlantica* (Kaal.) Müll.Frib., *Lophozia atlantica* (Kaal.) Schiffn.]. Sporadic. [1,7]; **I:** [15]; **II:** Kildin Island, Konstantinova [KPABG(H) 3964]; **IV:** [18]; **V:** Gremyakha-Vyrmes Area, Konstantinova [KPABG(H) 9580]; **VI**: [12,66]; **VIII**: [19,20]; **IX**: [19,20].

*Orthocaulis cavifolius* H.Buch & S.W.Arnell [*Sphenolobus cavifolius* (H.Buch & S.W.Arnell) Müll.Frib., *Anastrophyllum cavifolium* (H.Buch & S.W.Arnell) Lammes, *Lophozia cavifolia* (H.Buch & S.W.Arnell) R.M.Schust.]. Sporadic. EU: DD; **II:** Teriberka River Basin, Erieiv Mount, Schljakov [KPABG(H) 4419]; **VI:** [1,12,34,130].

*Pellia epiphylla* (L.) Corda [*Pellia borealis* Lorb. ex Müll.Frib. *Pellia epiphylla* var. *borealis* (Lorb. ex Müll.Frib.) Schljakov]. Sporadic. EU: LC; [1]; [69] {***KPABG(H)14114***}; **I:** [10]; **II:** Kharlov Isl., [1]; **IV:** [97]; **V**: [24,26,27]; **VII:** Kutsa Sanctuary, Konstantinova [KPABG(H) 1939]; **VIII**: [19,20]; **IX**: [19,20].

*Pellia neesiana* (Gottsche) Limpr. Frequent. EU: LC; [1,7]; [69] {***KPABG(H)19070***}. **I:** [15,65]; **II:** Teriberka Settlement, Borovichev [INEP 152672]; **III:** Lower Ponoj River, near Ponoj Village, Borovichev [INEP152579]; **IV**: [30]; **V**: [24,25,26,27]; **VI**: [12,17,33,35,66]; **VII:** Kutsa Sanctuary, Konstantinova [KPABG(H) 5083]; **VIII**: [19,20]; **IX**: [19,20].

*Peltolepis quadrata* (Saut.) Müll.Frib. [*Peltolepis grandis* (Lindb.) Lindb.]. Sporadic. EU: LC; MR: 3 (NT); [1]; **III:** [36,72]; **V:** [24,25,27,124]; **VI**: [17,18,33,34,39,84]; **VII**: [9].

*Plagiochila porelloides* (Torr. ex Nees) Lindenb. [*Plagiochila asplenioides* subsp. *porelloides* (Torr. ex Nees) Kaal.]. Sporadic. EU: LC; [7] (as *P. asplenioides*); [1]; **I:** [15]; **III:** Ponoj River Basin, Rusinga River, Schljakov [KPABG(H) 5061]; **IV**: [30]; **V**: [24,25,26,27]; **VI**: [13,17]; **VII:** [9]; as *P. asplenioides* s.lat.; [67]; **VIII**: [19,20]; **IX**: [19,20].

*Porella cordaeana* (Huebener) Moore [*Madotheca cordaeana* (Huebener) Dumort.]. Rare. EU: LC; [1]; **V:** [23,26,27]; **VII:** [9,67].

*Porella platyphylla* (L.) Pfeiff. [*Madotheca platyphylla* (L.) Dumort.]. Rare. EU: LC; MR: 3 (NT); [1,7]; **V:** [25,27]; **VII**: [9,46,67]; **IX**: [42].

*Prasanthus suecicus* (Gottsche) Lindb. Sporadic. EU: LC; MR: 3 (NT); [1,7]; **I**: [72,73]; **II:** [37,72,73]; **III**: [7], Ponoj River Basin, cape Orlov, Chilman [KPABG(H) 20661]; [36,72]; **V:** [27,73]; **VI:** [41,43,72]; **VIII:** [90].

*Protochilopsis grandiretis* (Lindb. ex Kaal.) A.V. Troitsky, Bakalin & Fedosov [*Schistochilopsis grandiretis* (Lindb. ex Kaal.) Konstant.; *Lophozia grandiretis* (Lindb. ex Kaal.) Schiffn., *Lophozia grandiretis* var. *proteidea* (Arnell) Arnell, *Massularia grandiretis* (Lindb. ex Kaal.) Schljakov]. Sporadic. EU: LC; [1]; **I:** [15]; **III:** [92]: Lower Ponoj River, Schljakov [KPABG(H)3449]; **IV:** [30]; **V**: [24,25,26,27,92]; **VI:** [12,13,17,66]; **VII:** [9,67]; **VIII:** [19,20]; **IX**: [19,20].

*Protolophozia elongata* (Steph.) Schljakov [*Lophozia elongata* Steph., *Orthocaulis elongatus* (Steph.) A.Evans]. Sporadic. EU: VU; RF: 2 (EN); MR: 3 (NT); [1]; [100] {***KPABG(H)9626***}; **II:** [1,32,72]; **III:** [1]; Lower Ponoj, Schljakov[KPABG(H) 3507]; [36]; **IV:** [18,46]; **V**: [25,27,31]; **VI:** [131]; **VIII:** [19,20,70]; **IX:** [19,20].

*Pseudolophozia debiliformis* (R.M.Schust. & Damsh.) Konstant. et Vilnet, *Lophozia debiliformis* R.M.Schust. & Damsh., *Lophozia debiliformis* var. *concolor* R.M.Schust. & Damsh., *Protolophozia debiliformis* (R.M.Schust. et Damsh.) Konstant.]. Sporadic. EU: NE; [100] {***KPABG(H)12772***}; **III:** [36]; **IV:** [81]; **V:** [24,27,81]; **VI:** [17]. {18}.

*Pseudomoerckia blyttii* (Mørch) Vilnet, Konstant., D.G.Long, N.D.Lockh. & Mamontov [*Moerckia blyttii* (Mørch) Brockm.]. Sporadic. EU: VU; [1,7,128]; [132] {***INEP500162***}; **I:** [65,77]; **II**: [1]; **IV**: [18,30]; **V**: [25,27,128].

*Ptilidium ciliare* (L.) Hampe. Common. EU: LC; [1,7]; **I:** [15]; **II:** Sem’ jstrovov, Breslina [KPABG(H)5563]; **III:** Lower Ponoj River, near Ponoj Village, Borovichev [INEP152576]; **IV**: [30]; **V**: [24,25,26,27]; **VI:** [12,17,33,66]; **VII:** [9]; **VIII**: [19,20]; **IX:** [19,20].

*Ptilidium pulcherrimum* (Weber) Vain. Common. EU: LC; [1,7]; **I:** [15]; **II:** Teriberka Settlement, Borovichev [INEP 152659]; **III:** Lower Ponoj River, near Ponoj Village, Borovichev [INEP152552]; **IV**: [30]; **V**: [24,25,26,27]; **VI**: [12,17,33,66]; **VII:** [9]; **VIII**: [19,20]; **IX**: [19,20].

*Radula complanata* (L.) Dumort. Sporadic. EU: LC; MR: 3 (NT); [1,7]; **II:** Teriberka Settlement, Borovichev [INEP 152669]; **III:** Rusinga River, Borovichev [INEP152559]; **IV**: [18,30]; **V**: [24,25,26,27]; **VII:** [9,67]; **VIII**: [19,20]; **IX**: [19,20].

*Radula lindenbergiana* Gottsche ex C.Hartm. Rare. EU: LC; [1]; **V**: [25,27]; **VII:** [9,67]; **VIII:** [70].

*Reboulia hemisphaerica* (L.) Raddi. Rare. EU: LC; [123] {***KPABG(H)18367***}; **V**: [24,27,28,124].

*Riccardia chamedryfolia* (With.) Grolle [*Riccardia sinuata* (Hook.) Trevis.]. Sporadic. EU: LC; [1]; **II:** Valley of Teriberka River, Mamontov [INEP400085]; **IV**: [18,30]; **V:** [24,25,27]; **VIII**: [19,20,70]; **IX:** [19,20].

*Riccardia incurvata* Lindb. Rare. EU: LC; MR: 3 (NT); [1]; **I:** [15,46]; **V:** [1]; **VII:** [1]; **IX:** Kovdozero Lake Basin (Tolvand), Konstantinova [KPABG(H) 14132].

*Riccardia latifrons* (Lindb.) Lindb. Frequent. EU: LC; [1,7]; **I:** [15]; **II:** Teriberka Settlement, Borovichev [INEP 152668]; **III:** Rusinga River, Borovichev [INEP 575]; **IV**: [30]; **V:** [24,25,26,27]; **VI:** [12,17]; **VIII**: [19,20]; **IX**: [19,20].

*Riccardia latifrons* subsp. *arctica* R.M. Schust. & Damsh. Rare. EU: NE; **IV**: [30].

*Riccardia multifida* (L.) Gray. Sporadic. EU: LC; [1]; **IV**: [30]; **V**: [1,26,27]; **VI**: [12,13].

*Riccardia palmata* (Hedw.) Carruth. Sporadic. EU: LC; MR: 3 (NT); **V**: [24,27,77]; **VII:** [72,74]; **VIII**: [18,72,90]; **IX:** Kovdozero Lake Basin, Konstantinova [KPABG(H) 14130]; [42].

*Riccia cavernosa* Hoffm. Rare. EU: LC; MR: 4 (DD); [1]; {***INEP500401: PQ605319***}; **IV**: [72,133]; **VII:** [9,67].

*Riccia fluitans* L. Rare. **VII:** Kutsa Sanctuary (Zakaznik), Hirvekallio, Tuomikoski [H]. {19}

*Riccia sorocarpa* Bisch. Rare. EU: LC; **VI**: [134].

*Ricciocarpos natans* (L.) Corda. Rare. EU: LC; **V**: Apatity Town [135].

Saccobasis polita (Nees) H.Buch [Sphenolobus politus (Nees) Steph., Tritomaria polita (Nees) Jørg., Tritomaria polita subsp. polymorpha R.M.Schust., Saccobasis polymorpha (R.M. Schust.) Schljakov]. Frequent. EU: LC; [1,7]; [121] {***KPABG(H) 6189, 8167, 8211, 8247, 125237, K20-3-12, CA13-19b***}; **I:** [15]; **II:** Valley of Teriberka River, Mamontov [INEP 400374]; **III:** Lower Ponoj River, Schljakov [KPABG(H):3413]; **IV**: [30]; **V**: [24,25,26,27]; **VI:** [12,13,17,33,35,66]; **VII:** [9,67]; **VIII**: [19,20]; **IX**: [19,20]. {20}

*Sauteria alpina* (Nees) Nees Sporadic. –EU: LC; MR: 3 (NT); [1,7,136]; **III:** [36,72]; **V:** [24,27,38,124]; **VI**: [13,33], 2020; **VII:** [9,67].

*Scapania aequiloba* (Schwägr.) Dumort. Rare. EU: LC; MR: 2 (VU); **III:** [36]; **V**: [1]; Valley of Imandra Lake, Schljakov [KPABG(H) 4475].

*Scapania apiculata* Spruce. Sporadic. EU: NT; MR: 3 (NT); [1]; **IV**: [30,46]; **V**: [24,27]; **VII:** [1,9,46,67]; **VIII:** [81]; **IX**: [42].

*Scapania calcicola* (Arnell & J.Perss.) Ingham. Rare. EU: LC; **V**: [137]. Borovichev {KPABG(H) 21365].

*Scapania crassiretis* Bryhn. Sporadic. EU: LC; [1]; [127] {***KPABG**(H**)123534***}; [100] {***KPABG**(H**)8074***}; **III:** Lumbovskiy Bay, Borovichev [KPABG(H) 18653]; **IV**: [30]; **V**: [25,27]; **VI**: [12,13,33,66].

*Scapania curta* (Mart.) Dumort. Frequent. EU: LC; [1,7]; [100] {***KPABG**(H**)8104*}**; **I**: [78]; **II:** Valley of Teriberka River, Mamontov [INEP 400380]; **III:** Lower Ponoj River, Schljakov [KPABG(H) 4480]; **IV**: [30]; **V**: [24,25,26,27]; **VI**: [12,17,66]; **VII:** [9,67]; **VIII**: [19,20]; **IX**: Kovdozero Lake Basin (Tolvand), Konstantinova [KPABG(H) 14127].

*Scapania curta* var. *grandiretis* R.M.Schust. Rare. EU: NE; **VI**: [66].

*Scapania cuspiduligera* (Nees) Müll.Frib. Sporadic. EU: LC; [1]; **I**: [13,78]; **IV**: [30]; **V**: [25,26,27,68]; **VI:** [12]; Khibiny Mts., Konstantinova [KPABG(H)18362]; [33]; **VII**: [1,67]; **IX**: [19].

*Scapania degenii* Schiffn. ex Müll.Frib. [*Scapania degenii* var. *dubia* R.M.Schust.]. Rare. EU: NT; **VI**: [17,18].

*Scapania gymnostomophila* Kaal. Sporadic. EU: LC; [1]; [100] {***KPABG(H)6913***}; **I:** [1]; **IV**: [30]; **V:** [25,27,68]; **VI**: [84]; **VII:** [1,9,67]; **IX:** [18].

*Scapania hyperborea* Jørg. Frequent. EU: LC; Arnell. 1956; [1]; [100] {***KPABG(H)12263***}; **I:** [15]; **II:** Valley of Teriberka River, Mamontov [INEP 400057]; **III:** Lower Ponoj River, Schljakov [KPABG(H) 4510]; **IV:** [30]; **V**: [24,26,27]; **VI:** [12,17,33,66]; **VII:** Kutsa Sanctuary, Konstantinova [KPABG(H)3969]; **VIII:** [19,20]; **IX**: [19,20].

*Scapania irrigua* (Nees) Nees. Common. EU: LC; [1,7]; [100] {***KPABG(H)9587***}; **I:** [15]; **II:** Iokan’ga River Basin, Schljakov [KPABG(H) 4777; **III:** Lower Ponoj River, Schljakov [KPABG(H) 4642]; **IV**: [30]; **V**: [24,25,26,27]; **VI:** [12,17,33,66]; **VII:** Kutsa Sanctuary, valley of Nivayarvi Lake, Konstantinova, [KPABG(H)2094]; **VIII:** [19,20]; **IX**: [19,20].

*Scapania kaurinii* Ryan. Sporadic. EU: VU; MR 3 (NT); [1]; **I**: [1]; **II:** Dal’nye Zelentsy village, Schljakov [KPABG(H)4590]; **V**: [26,27]; **VI**: [1,40,72,89].

*Scapania lingulata* H.Buch. Sporadic EU: NT; **I**: [78]; **V:** [27,81]; **VI:** Khibiny Mt., Konstantinova, [KPABG12278]; **VIII**: [81].

*Scapania mucronata* H.Buch. Sporadic. EU: LC; [1]; **I:** [15]; **V:** [24,25,26,27]; **VI**: [12,17,33]; **VII:** [9,67]; **VIII**: [90]; **IX:** Kanda River valley, Konstantinova [KPABG(H) 4593].

*Scapania obcordata* (Berggr.) S.W.Arnell [*Scapania paradoxa* R.M.Schust., *Scapania lapponica* (Arnell et C.E.O.Jensen) Steph.]. Sporadic. EU: LC; [1]; **I:** [15]; **II:** Iokan’ga River Basin, Konstantinova [KPABG(H) 6153]; **III:** Lower Ponoj River, Schljakov [KPABG(H) 4610]; **IV**: [30]; **V:** [27]; **VI:** [12,17,66,92]; **VII:** Tumcha River Valley, Schljakov [KPABG(H) 4619] **VIII**: [19,20]; **IX**: [19,20].

*Scapania obscura* (Arnell & C.E.O.Jensen) Schiffn. Rare. EU: DD; [1]; {***KPABG(H)8826: PQ605320***}. **I**: [10,15]; **V**: [24,27].

*Scapania paludicola* Loeske & Müll.Frib. [*Scapania paludicola* var. *rufescens* Damsh. nom. inval.]. Frequent. EU: LC; [7]; [1]; **I:** [15]; **II:** Iokan’ga River Basin, Sukhaya River, Konstantinova [KPABG(H)6169]; **III:** Lower Ponoj River, Schljakov [KPABG(H)4621]; **IV**: [30]; **V:** [24,25,26,27]; **VI:** [12,17,66]; **VII:** [9,67]; **VIII:** [19,20]; **IX**: [19,20].

*Scapania paludosa* (Müll.Frib.) Müll.Frib. [*Scapania paludosa* var. *isoloba* Müll.Frib., *Scapania paludosa* var. *rubiginosa* Müll.Frib., *Scapania paludosa* var. *vogesiaca* Müll.Frib.]. Frequent. EU: LC; [1]; **I:** [15]; **II:** Kildin Island, Konstantinova [KPABG(H) 4647]; **IV:** [18]; **V**: [24,25,27]; **VI:** [12,13,17,35]; **IX:** [18].

*Scapania parvifolia* Warnst. Sporadic. EU: NT; **I**: [78]; **IV:** [97]; **V:** [24,25,27]; **VI:** [12]; **VII:** [9,67]; **IX:** Kovdozero Lake basin, Konstantinova [KPABG(H) 14136].

*Scapania praetervisa* Meyl. [*Scapania mucronata* subsp. *praetervisa* (Meyl.) R.M.Schust., *Scapania mucronata* var. *arvernica* (Culm.) Müll.Frib.] Sporadic. EU: LC; [1,7]; **I:** [1]; **II:** Iokan’ga River Basin, Sukhaya River, Konstantinova [KPABG(H) 6148]; **III:** [1]; **IV**: [30]; **V**: [1,25,26,27]; **VI:** [66]; **VII:** Kutsa Sanctuary, Konstantinova [KPABG(H) 4943]; **VIII:** [19,20]; **IX**: Kanda River valley at the base of Gremyashka Mt., Konstantinova, [KPABG(H) 6124]; Mikkov Isl., Borovichev [INEP153679].

*Scapania scandica* (Arnell & H.Buch) Macvicar. Sporadic. EU: LC; [1]; **II**: Hill on the outskirts of the city of Murmansk, Konstantinova [KPABG(H) 8086]; **IV**: [30]; **V**: [24,25,26,27]; **VI:** [17,66]; **VII:** Kutsa Sanctuary, Konstantinova [KPABG(H) 6122]; **VIII**: [19,20]; **IX:** [19,20].

*Scapania scandica* var. *grandiretis* (Schljakov) Schljakov [*Scapania parvifolia* Warnst. var. *grandiretis* Schljakov] –. EU: NE; **III**: [138]. {21}

*Scapania simmonsii* Bryhn & Kaal. Rare. EU: VU; MR: 2 (VU); **I:** [1,7]; [100] {***KPABG(H)6677***}; **V**: [23,24,27]; **VI:** [17,18,84].

*Scapania sphaerifera* H. Buch & Tuom. Rare. EU: CR; RF: 3 (NT); MR: 1a (CR); **VII**: [1,9,41,46,67]. {***KPABG(H)****** Ku-10-5-20: PQ765530, PQ767096***}.

*Scapania spitsbergensis* (Lindb.) Müll.Frib. Sporadic. EU: VU; MR: 3 (NT); [1]; **III:** [86]; **IV**: [18]; **V**: [25,27,68]; **VII:** [9,46,67]. {22}

*Scapania subalpina* (Nees ex Lindenb.) Dumort. Common. EU: LC; [1]; [121] {***KPABG(H) NK 28.09.2019***}; **I:** [15]; **II:** Kildin Island[KPABG(H) 4887]; **III:** Lower Ponoj River, Schljakov [KPABG(H) 4083]; **IV**: [30]; **V**: [24,25,26,27]; **VI:** [12,17,33,66]; **VII:** [9,67]; **VIII**: [19,20]; **IX**: [19,20].

*Scapania tundrae* (Arnell) H.Buch. Sporadic. EU: LC; MR: 3 (NT); **II:** [1]; **V:** [25,27]; **VII:** near Kuolojarvi, Salla-tunturi, Schljakov [KPABG(H) 20191]; **VIII:** [1].

*Scapania uliginosa* (Lindenb.) Dumort. Common. EU: LC; [1,7]; [100] {***KPABG(H)8849***}; [121] {***KPABG(H) NK 26-3-88***}; **I:** [15]; **II:** Iokan’ga River Basin, valley of Sukhaya River, Konstantinova, [KPABG(H) 6201]; **IV**: [30]; **V**: [24,25,26,27]; **VI**: [12,17,33,66]; **VII:** [9]; **IX:** Kanda River Valley Konstantinova, [KPABG(H) 4788].

*Scapania umbrosa* (Schrad.) Dumort. Sporadic. EU: LC; MR: 3; [1]; **IV**: [30,46]; **V**: [26,27,68,73]; **VI:** [43]; **VIII:** [19,20]; **IX**: Mikkov Isl., Borovichev [INEP153688].

*Scapania undulata* (L.) Dumort. Frequent. EU: LC; [1,7]; [100] {***KPABG(H)9554***}; **I:** [15]; **II**: Iokan’ga River Basin, Sukhaya River, Konstantinova [KPABG(H) 6201]; **III**: Lower Ponoj River, Dombrovskaya [KPABG(H) 4836]; **IV**: [30]; **V:** [24,25,26,27]; **VI:** [12,17,66]; **VII:** [9,67]; **VIII**: [19,20]; **IX**: [19,20].

*Schistochilopsis incisa* (Schrad.) Konstant. [*Lophozia incisa* (Schrad.) Dumort., *Massularia incisa* (Schrad.) Schljakov, *Schistochilopsis incisa* var. *inermis* (Müll.Frib.) Konstant.]. Frequent. EU: LC; [1,7]; [100] {***KPABG(H)9499***}; **I:** [15]; **II:** Iokan’ga River Basin, valley of Sukhaya River, Konstantinova [KPABG(H)6074]; **III:** Lower Ponoj River, Schljakov [KPABG(H) 5201]; **IV**: [30]; **V:** [24,25,26,27]; **VI**: [12,17,33,66]; **VII:** [9,67]; **VIII**: [19,20]; **IX**: [19,20].

*Schistochilopsis opacifolia* (Culm. ex Meyl.) Konstant. [*Lophozia opacifolia* Culm. ex Meyl., *Lophozia incisa* subsp. *opacifolia* (Culm. ex Meyl.) R.M.Schust. & Damsh., *Massularia opacifolia* (Culm. ex Meyl.) Schljakov]. Sporadic. EU: LC; [1]; [123] {***KPABG(H)8205***}; **I:** [15]; **II:** Iokan’ga River Basin, valley of Zolotaya River, Konstantinova [KPABG(H)6157]; **III:** Lower Ponoj River, Schljakov [KPABG(H) 5201]; **IV:** [30]; **V**: [24,25,27]; **VI:** [12,17,33,66].

*Schizophyllopsis sphenoloboides* (R.M.Schust.) Váňa & L.Söderstr. [*Anastrophyllum sphenoloboides* R.M. Schust.]. Rare. EU: EN; MR: 2 (VU); [1]; [100] {***KPABG(H)8831***}; **II**: [37,72]; **III**: [86]; **V**: [1,24,27,72]; **VI**: [72,92].

*Schljakovia kunzeana* (Huebener) Konstant. & Vilnet [*Barbilophozia kunzeana* (Huebener) Müll.Frib., *Lophozia kunzeana* (Huebener) A.Evans, *Orthocaulis kunzeanus* (Huebener) H.Buch]. Common.– EU: LC; [1,7]; [127] {***KPABG(H)9486***]; [***KPABG(H)12257; PQ585839, PQ605321***}; **I**: [15]; **II**: Iokan’ga River Basin, valley of Sukhaya River, Konstantinova [KPABG 6169]; **III**: Lower Ponoj River, Schljakov [KPABG(H) 3565]; **IV**: [30]; **V**: [24,25,27]; **VI**: [12,17,33,35,66]; **VII**: [9,67]; **VIII**: [19,20]; **IX**: [19,20].

*Schljakovianthus quadrilobus* (Lindb.) Konstant. & Vilnet [*Barbilophozia quadriloba* (Lindb.) Loeske, *Lophozia quadriloba* (Lindb.) A.Evans, *Lophozia quadriloba* var. *glareosa* (Jørg.) Bryhn et Kaal., *Orthocaulis quadrilobus* (Lindb.) H.Buch]. Frequent. EU: LC; [1,7]; **I:** [15]; **III**: Lower Ponoj River, Orlov Cape, Schljakov [KPABG(H) 4081]; **IV**: [30]; **V:** [24,25,26,27]; **VI**: [12,17,33,66]; **VII:** [9,67]; Kutsa Sanctuary, Konstantinova [KPABG(H)2437]; **VIII:** Umba River Valley, Konstantinova [KPABG(H)121733]; **IX**: Mikkov Isl., Borovichev [INEP153685].

*Solenostoma confertissimum* (Nees) Schljakov [*Jungermannia confertissima* Nees, *Solenostoma levieri* (Steph.) Steph.]. Sporadic. EU: LC; [1]; [139] {***KPABG(H)8246***}; {***KPABG(H)9642: PQ605322***}; **I:** [15]; **II:** Valley of Teriberka River, Andrejeva [KPABG(H) 1867]; **III:** [1]; **IV**: [18]; **V**: [24,25,26,27]; **VI**: [13,17,66]; **VIII:** [1]; **IX**: Mikkov Isl., Borovichev [INEP153687].

*Solenostoma hyalinum* (Lyell) Mitt. [*Eucalyx hyalina* (Lyell) F.Lees, *Jungermannia hyalina* Lyell, *Plectocolea hyalina* (Lyell) Mitt.]. Sporadic. EU: LC; [1,7]; [139] {***KPABG(H)8165***]; [***KPABG(H)2480: PQ605323, PQ605328; KPABG(H)6772: PQ605325, PQ605329***}; **I:** [15]; **II:** Iokan’ga River Basin, valley of Salmon Lake, Konstantinova [KPABG 6197]; **V:** [24,26,27]; **VI**: [12,17,33,66]; **VIII:** [19,20]; **IX**: [19,20]; Kovdozero Lake Basin, Tolvand Lake Basin, Konstantinova [KPABG 14139].

*Solenostoma obovatum* (Nees) C.Massal. [*Eucalyx obovata* (Nees.) F.Lees, *Jungermannia obovata* Nees, *Plectocolea obovata* (Nees) Mitt., *Plectocolea subelliptica* (Lindb. ex Heeg) A.Evans]. Frequent.– EU: LC; [1]; [139] {***KPABG(H)9543***}; [140] {***KPABG(H)6176***}; **I:** [15]; **II:** Kildin Isl., Konstantinova [KPABG 4977]; **III:** Lower Ponoj River, near Ponoj Village, Schljakov [KPABG(H)4257]; **IV:** [30]; **V**: [24,27]; **VI**: [13]; Konstantinova,1976, 2001b; [17,33,35]; **VII:** [67]; **VIII**: [19,20]; **IX:** [19,20].

*Solenostoma pusillum* (C.E.O.Jensen) Steph. [*Aplozia pusilla* C.E.O.Jensen, *Jungermannia pusilla* (C.E.O.Jensen) H.Buch, *Jungermannia jenseniana* Grolle]. Rare. EU: NE; [1]; **I:** [15]; **III:** [1]; **VI:** [13,17]. {23}

*Solenostoma sphaerocarpum* (Hook.) Steph. [*Jungermannia sphaerocarpa* Hook., *Jungermannia sphaerocarpa* var. *nana* (Nees ex Flot.) Frye & L.Clark nom. illeg., *Solenostoma sphaerocarpum* var. *nanum* (Nees ex Flot.) R.M.Schust.]. Frequent. EU: LC; [1]; [139] {***KPABG(H)8737***}; **I:** [15]; **II:** Teriberka River Basin, Mamontov [INEP400381]; **III:** Lower Ponoj River, near Ponoj Village, Schljakov [KPABG(H) 5202]; **IV:** [30]; **V:** [24,25,26,27]; **VI**: [12,17,33,66]; **VII:** [9,67]; **VIII:** Ponoj River Basin, near Krasnoshshelie Village, Schljakov [KPABG(H) 4241]; **IX**: Mikkov Isl., Borovichev [INEP153686].

*Sphenolobus minutus* (Schreb.) Berggr. [*Anastrophyllum minutum* (Schreb.) R.M.Schust., *Anastrophyllum minutum* var. weberi (Mart.) Kartt.]. Common. [1,7]; {***KPABG(H)126010: PQ765531, PQ767097; KPABG(H)126021: PQ765532, PQ767098, PQ767095***}; EU: LC; **I:** [15]; **II:** Iokan’ga River Basin, valley of Sukhaya River, Konstantinova [KPABG(H) 6153]; **III:** Lumbovka Bay, Borovichev [KPABG(H) 18647]; **IV**: [30]; **V**: [24,25,26,27] 14b; **VI**: [12,13,17,33,66]; **VII:** [9,67]; **VIII:** [19,20]; **IX**: [19,20].

*Sphenolobus saxicola* (Schrad.) Steph. [*Anastrophyllum saxicola* (Schrad.) R.M.Schust.]. Sporadic. EU: LC; [1,7]; **IV:** [30]; **V**: [24,25,26,27]; **VI:** [12,66]; **VII**: [1,9,67]; **IX:** Kovdozero Lake Valley, Konstantinova [KPABG(H) 14122].

*Tetralophozia setiformis* (Ehrh.) Schljakov. Frequent. EU: LC; [1,7]; [141] {***KPABG(H)18022***}; **I:** [15]; **II:** Severomorsk, Dombrovskaya [KPABG(H) 2281]; **III:** Rusinga River, Borovichev [INEP152562]; **IV**: [30]; **V**: [24,25,26,27,46]; **VI**: [12,17,33,66]; **VII:** [9,67]; **VIII:** near Umba village, Borovichev [KPABG(H) 20739]; **IX**: [19,20].

*Trilophozia quinquedentata* (Huds.) Bakalin [*Tritomaria quinquedentata* (Huds.) H.Buch, *Tritomaria quinquedentata* subsp. *turgida* (Lindb.) Damsh., *Trilophozia quinquedentata* var. *turgida* (Lindb.) Konstant., *Tritomaria quinquedentata* f. *gracilis* R.M.Schust., *Tritomaria quinquedentata* var. *dentata* S.W.Arnell nom. inval., *Tritomaria quinquedentata* var. *grandiretis* H.Buch & S.W.Arnell]. Common. EU: LC; [1,7]; {***KPABG(H)2819: PQ765533, PQ767099***}; **I:** [15]; **III:** Lower Ponoj River, near Ponoj Village, Schljakov [KPABG(H) 3986]; **IV**: [30]; **V**: [24,25,26,27,28]; **VI**: [12,17,33,35,66]; **VII:** [9,67]; **VIII**: [19,20]; **IX**: [19,20].

*Tritomaria exsectiformis* (Breidl.) Schiffn. ex Loeske [*Sphenolobus exsectiformis* (Breidl.) Steph.]. Sporadic. EU: LC; MR: 3 (NT); [1]; **V:** [23,25,27]; **VI**: [72]; **VII**: [1,9,46,67]; **VIII**: [1,19,72,81,90]; **IX**: [42].

*Tritomaria scitula* (Taylor) Jørg. [*Sphenolobus exsectiformis* var. *aequilobus* (Culm.) Müll.Frib.]. Sporadic. EU: LC; [1,7]; **I:** [15]; **II:** Teriberka Settlement, Borovichev [INEP 152667]; **III:** Lower Ponoj River, near Ponoj Village, Schljakov [KPABG(H)3449]; **IV**: [30]; **V**: [24,27]; **VI:** [12,13,33]; **VII:** [9,67].

### 2.2. Excluded Taxa

*Apopellia endiviifolia* (Dicks.) Nebel & D.Quandt [*Pellia endiviifolia* (Dicks.) Dumort.], [19,70]; was reidentified as *Apopellia alpicola* [69].

*Harpanthus scutatus* (F. Weber et D. Mohr) Spruce. The species was recorded for the Lapland State Nature Reserve [11], but no supporting specimen could be found, and given the distribution and ecology of the species, this record is clearly erroneous.

*Lophozia rufescens* Schljakov. This taxon is treated as a synonym of *Barbilophozia sudetica* (Nees ex Huebener) L.Söderstr., De Roo et Hedd. [22].

*Marsupella spiniloba* R.M. Schust. et Damsh. The species was recorded for Khibiny Mts. [66] however, was reidentified as *Marsupella sprucei* (Limpr.) Bernet. *M. spiniloba* is a poorly known and problematic taxon described from Greenland [142] and so far also known from single localities in the Scandinavian countries and Alaska [108].

*Moerckia hibernica* (Hook.) Gottsche. As it was recently shown [128], *Moerckia hibernica* is not synonymous with *M. flotoviana* (Nees) Schiffn. *M. hibernica* has a very limited distribution (mainly in the British Isles), and all specimens identified as *M. hibernica* from the Murmansk Region belong to *M. flotoviana.*

*Plagiochila arctica* Bryhn & Kaal. [*Plagiochila asplenioides* subsp. *arctica* (Bryhn & Kaal.) R.M.Schust.] According [1] all previous records [12,13,92] are misinterpretations of mire forms of *Plagiochila porelloides* (Torr. ex Nees) Lindenb.

*Saccobasis polymorpha* (R.M. Schust.) Schljakov This taxon has recently been synonymized with *Saccobasis polita* (Nees) H.Buch [121].

*Scapania paradoxa* R.M. Schust. The species was recorded by [19]. This taxon recently is treated as *Scapania obcordata* (Berggr.) S.W.Arnell [59].

*Sphenolobopsis pearsonii* (Spruce) R.M.Schust. [*Cephaloziella pearsonii* (Spruce.) Douin.] The species was recorded for Khibiny Mts. [1] and Lovozero Mts. [17]; but later all specimens were revised and referred to *Eremonotus myriocarpus* (Carrington) Lindb. et Kaal. ex Pearson [94].

*Solenostoma gracillimum* (Sm.) R.M.Schust. [*Jungermannia gracillima* Sm., *Jungermannia gracillima* var. *crenulata* (Mitt.) Damsh., *Jungermannia crenulata* Sm. l.c. *Aplozia crenulata* (Mitt.) Dumort.] Record of [143] based on Ångström was excluded by [1].

*Solenostoma subellipticum* (Lindb. ex Heeg) R.M.Schust. [*Jungermannia subelliptica* (Lindb. ex Heeg) Levier, *Plectocolea subelliptica* (Lindb. ex Heeg) A.Evans, *Jungermannia obovata* subsp. *minor* (Carrington) Damsh.] is recorded for Murmansk Region [24,26,68] however here we follow although with some doubt most recent authors and consider the taxon as a synonym of *S. obovatum.*

### 2.3. Comments

1. *Barbilophozia sudetica* var. *anomala* (Schljakov) Konstant. var. nova—*Lophozia alpestris* (F. Weber) A. Evans var. *anomala* Sch1jakov, *Novosti sistematiki nizshikh rastenii,* 11, 1974: 352 *Lophozia sudetica* (Nees) Grolle var. *anomala* (Sch1jakov) Schljakov comb. nov. Basionym: Lophozia alpestris var. anomala *Novosti sistematiki nizshikh rastenii*, 12, 1976: 228. Schljakov #7a isotype [KPABG(H)19169] Schljakov#208 paratype [KPABG(H)19168]. In the *World Checklist of Hornworts and Liverworts* [144] and in the European checklist [59], this taxon is synonymized with *B. sudetica*. The taxonomical status of this variety is not clear to us and we prefer to treat it as a variety until a special study of the type material or material from the type locality has been made.

2. *Blepharostoma brevirete* (Bryhn & Kaal.) Vilnet & Bakalin is mostly considered a variety or subspecies of *B. trichophyllum* (L.) Dumort. [1,59,144,145]. The species s.lat. is easily recognizable, so it is not always examined under a microscope, without which it is almost impossible to distinguish specimens as *B. brevirete* or *B. trichophyllum*. Therefore, to clarify the distribution of these two species in the region, the revision of all specimens is necessary. This is especially true for other recently described species, two of which are listed for the Murmansk Region only on the basis of a molecular genetic study [80].

3. *Calypogeia trichomanis* (L.) Corda was recorded for the Murmansk Region by Savicz and [8] without an exact locality and by [11] as from the Lapland State Nature Reserve. Judging by the description of species in the first cited publication, this record refers to *C. muelleriana.*

4. The distribution of species of the genus *Cephaloziella* in the Murmansk Region needs to be clarified based on a global revision of the genus.

5. *Fossombronia wondraczekii* (Corda) Dumort. ex Lindb. was recorded by [75] based on a specimen collected by Chilman in 1892 in Khibiny Mts. stored in H and KPABG.

6. *Fuscocephaloziopsis affinis* (Lindb. ex Steph.) Váňa & L.Söderstr. has been recorded for Kem’-Ludy Archipelago (in the north of the Republic of Karelia) [19]. In the Murmansk Region and elsewhere in Russia, this taxon is considered a synonym of *Fuscocephaloziopsis lunulifolia* (Dumort.) Váňa & L.Söderstr., following [145]. A revision of *F. lunulifolia* specimens is necessary to reveal the distribution of the taxon in the region.

7. *Cephalozia leucantha* var. *robusta* Schljakov was described from Khibiny Mt. [98]. The variety has not been recognized recently and has been treated as synonym of *Cephalozia leucantha* Spruce [59]. We do not have a definite position on this taxon, and so far we are following the treatment of European authors (l.c.).

8. *Jungermannia atrovirens* Dumort.—[1] doubted the correctness of the records of the species for the Murmansk Region. However, J. Váňa, who studied several specimens from the Murmansk region, confirmed records of the species in the Murmansk Region. The genus *Jungermannia* is genetically very variable. An integrative revision of specimens from the Murmansk region is needed for confirmation of the records of this species.

9. The differentiation of *Jungermannia polaris* Lindb. and *J. pumila* With. in the Arctic presents great difficulties. According to [112,146]’s diversification of *J. polaris* at the species level is not complete and they should be considered as subspecies. Apparently, all populations in the Murmansk Region belong to *J. polaris*, but we prefer to leave it at the species level until special studies are carried out. A distribution of *Jungermannia polaris* and *J. pumila* in the Murmansk Region is given based on publications and herbarium labels.

10. The interpretation of the *Lophozia* species is very confusing. There was no revision of the specimens stored in the herbarium during this study, and here we cite the distribution of *Lophozia* species in the different provinces as they have been given in publications realizing, however, that this is mainly a reflection of diverse, inconsistent interpretations of species by different authors. These are mainly interpretations accepted in Russia and the Murmansk Region in particular [22,108,114].

11. *Lophozia ventricosa* (Dicks.) Dumort. auct.: at different times, the interpretations of the species in the territory of the region have differed; here, we refrain from specifying the distribution of the species according to the floristic regions.

12. *Marchantia polymorpha* L.: the distribution of subspecies needs to be clarified, since, most often, the species is treated in a broad sense, without division into subspecies.

13. *Marsupella subemarginata*: Bakalin & Fedosov, recently described from Kamchatka [147], has also been recorded in Switzerland, Japan (l.c.), Chukotka [148], the Czech Republic [149], and the Polish and Slovak Tatras [150]. The species is morphologically almost indistinguishable from *M. emarginata* Reinw., Blume & Nees. Given the known distribution of the species, its presence in the Murmansk region is quite likely. Integrative study of specimens collected in the Murmansk Region is needed to clarify the presence or absence of this taxon in the region.

14. In addition to the two varieties recently accepted—*Mesoptychia heterocolpos* var. *arctica* (S.W.Arnell) L.Söderstr. & Váňa and *M. heterocolpos* var. *harpanthoides* (Bryhn & Kaal.) L.Söderstr. & Váňa—another one (*Leiocolea heterocolpos* var. *savicziae* [138]) has been described. However, it was later synonymized with *Mesoptychia heterocolpos* var. *arctica* [114]. The distribution in the region of both accepted varieties needs to be studied. According to [1], var. *arctica* sporadically occurs in all provinces, whereas var. *harpanthoides* has only been recorded for the lower Rusinga River. In other publications on the Murmansk region, these varieties were not considered.

15. It has recently been shown that all records of *Moerckia hibernica* (Hook.) Rabenh. for the Murmansk Region should be referred to as *M. flotoviana* (Nees) Schiffn.

16. The morphologically quite similar *Nardia pacifica* Bakalin occurs in Karelia [151]. Given that this species has been recently described (l.c.) and is little known, it is likely that it can also occur in the Murmansk region. A revision of specimens of *Nardia japonica* Steph. based on an integrative approach is needed.

17. It has recently been shown that *Rudolgaea fascinifera* (Potemkin) Potemkin & Vilnet can be misidentified as *Obtusifolium obtusum* (Lindb.) S.W.Arnell [152]. It is necessary to revise the specimens collected in rich fens in the Murmansk Province and identified as *Obtusifolium obtusum*.

18. *Pseudolophozia debiliformis* (R.M.Schust. & Damsh.) Konstant. & Vilnet has recently been treated as a synonym of *Barbilophozia sudetica* (Nees ex Huebener) L.Söderstr., De Roo & Hedd. [59,144], but we prefer to keep it as a separate species before careful study of specimens from different regions, including of the type locality.

19. The record of *Riccia fluitans* L. is based on a specimen collected by R. Tuomikoski in 1934 in Hirvekallio Rocks (Kutsa Sanctuary (Zakaznik) [H]) studied by N. Konstantinova.

20. Following its recent treatment [121], we do not consider *Saccobasis polymorpha* (R.M.Schust.) Schljakov a separate species.

21. *Scapania scandica* var. *grandiretis* (Schljakov) Schljakov was described by [138] as *Scapania parvifolia* Warnst. var. *grandiretis* Schljakov from the Lower Ponoj River, based on a specimen collected by A. V. Dombrovskaja [KPABG(H)19171, *holotype*].

22. *Scapania spitsbergensis* was recorded for the Lovozerskie Mts [17], but the specimens was reidentified by Dr. Seung-Se Choi.

23. *Solenostoma pusillum* (C.E.O.Jensen) Steph. has recently been treated as a synonym of *Solenostoma sphaerocarpum* (Hook.) Steph. by some authors [59]; however, we prefer to consider this taxon a separate species for now.

## 3. Discussion

### 3.1. Evaluation of Diversity of Liverworts in Murmansk Region

A total of 210 species have been found in the Murmansk Region. This is one of the richest and most well-studied territories in terms of the diversity of liverworts in Russia. The diversity of liverworts is noticeably greater than in the Karelia Republic (187 species accordingly [153]), bordering the Murmansk Region in the south and exceeding it in area by about 1.2 times, and in length from north to south by one and a half times. Approximately the same number of species (205) are given for the Magadan Region [154], the northern regions of which are located at approximately the same latitudes as the Murmansk Region, but they significantly (by slightly more than three times) exceed the Murmansk Region in area and length (south to 58°55′ N), and the height of the mountains are more than 2000 m high, as against the highest point of 1200 m in the Murmansk Region. The number of species found in the Murmansk Region is only slightly less than in Kamchatka Peninsula and adjacent islands, for which 226 species are listed [155]. This is despite the fact that the area of Kamchatka (about 255,000 km^2^) exceeds the area of the Murmansk Region by almost two times, and the absolute heights in Kamchatka Peninsula reach 4700 m above sea level in Klyuchevskoy volcano [155] versus the maximum height in the Murmansk Region of 1200 m. However, it should be borne in mind that both the Magadan Region and Kamchatka Peninsula are much less studied areas compared to the Murmansk Region.

The number of liverworts in the countries bordering the Murmansk Region in the west is higher than in the region. Finland is the most similar in area, natural conditions and the number of known species ca.240 [155]). Nevertheless, taking into account the almost 2.5 times larger area of the country and the fact that a significant part of it is located much further south, the excess of the number of species of 20 does not seem high. The species recorded in Finland and not found in the Murmansk Region are mostly species with a more southern, temperate distribution (*Anastrophyllum michauxii, Bazzania tricrenata, B. trilobata, Calypogeia fissa*, *Cephalozia catenulata, C. macounii, Cephaloziella massalongi, C. stellulifera, Cololejeunea calcarea, Douinia ovata, Fossombronia fleischeri, Frullania dilatata, F. fragilifolia, Harpanthus scutatus, Heterogemma capitata, Lejeunea patens, Lepidozia cupressina, Lophocolea bidentata, L. coadunata, Lunularia cruciata*, *Mannia fragrans, M. sibirica, Marsupella funckii, Nardia compressa, Odontoschisma denudatum, Plagiochila asplenioides, Riccia beyrichiana, Riccia bifurca, R. canaliculata, R. ciliata, R. glauca, R. huebeneriana, R. rhenana, R. warnstorfii, Scapania carinthiaca, S. compacta, S. nemorea, Syzygiella autumnalis*, *Trichocolea tomentella*). The exception is *Rudolgaea borealis,* which occurs in a certain type of swamps and which is likely to be found in the future study of the liverworts of the Murmansk Region. The diversity of liverworts in Norway and Sweden, two other closely located and fairly well-studied countries, is noticeably higher, at about300 and 275 species, respectively. This is due to the significantly large size of the territories of these countries, the significant length from north to south, and the high mountains, with heights up to 2106 m in Sweden and 2469 m in Norway.

### 3.2. Rare Species

Rare species in the Murmansk Region species, that is, those that are known from just one to five localities, comprise 50 species, or 25% of the species known in the region. Four of them are recently described species *(Apopellia alpicola, A. megaspora, Blepharostoma neglectum, B. primum, Frullania subarctica*). All these species are described on the basis of an integrative approach [69,80,96] using molecular genetic methods. The identification of these species on the basis of morphological criteria alone is very difficult, especially for species of the genus *Blepharostoma.* A large group of rare liverworts consists of obligate calciphiles, including both recently found species (*Asterella lindenbergiana, Lophoziopsis pellucida, Mannia pilosa, Mannia triandra, Oleolophozia perssonii, Reboulia hemisphaerica, Scapania calcicola)* and the species already mentioned by Finnish and Swedish bryologists, e.g., Arnellia *fennica*, *Scapania simmonsii* [7]. The largest group among the newly recorded species are species which generally have a more southern or northern distribution, located in the Murmansk Region near the limit of their distribution. Among them are species at the northern limit of their distribution, e.g., Lejeunea *cavifolia, Nowellia curvifolia, Porella cordaeana, P. platyphylla, Radula lindenbergiana*. All recently found annual species (*Fossombronia foveolata, F. incurva, F. wondraczekii, Riccia cavernosa, R. sorocarpa, Ricciocarpos natans*) can be attributed to this group. A somewhat smaller group consists of species with a predominantly Arctic distribution, including *Calycularia laxa, Lophoziopsis polaris, Schizophyllopsis sphenoloboides, Lophoziopsis excisa* var. *elegans, L. rubrigemma.* A significant number of the newly found species are very small and poorly known *(Cephaloziella arctogena, C. integerrima, C. polystratosa, Isopaches alboviridis, I. decolorans, Diplophyllum obtusifolium, Solenostoma pusillum*) or just poorly known liverworts like *Lophozia silvicoloides* and *Lophozia schusteriana.*

About half of the liverworts recorded in the region are sporadically occurring species. The vast majority of them are stenotopic species confined to sporadically occurring habitats in the region. This group includes a majority of non-obligate calciphiles, as well as many epixylic species, either occurring in the Murmansk Region at the northern border of their distribution, or as species with a predominantly Arctic distribution. About 30% of species known in the region are widespread (common) and frequently occurring species. These are species restricted to rocks and fine-earth widely distributed in the area, as well as boggy areas. Almost all of them are found in all biogeographical provinces. The absence of some in one or two provinces is due to the poor study of these areas.

The frequency of occurrence of the species in the region is estimated both on the basis of the number of specimens in herbariums and on the basis of our experience in collecting in the region and field notes. However, it is obvious that there may be certain distortions due to various reasons, ranging from the abundance of the species, its size, and its proximity to certain habitats. In particular, the greatest attention in the course of studying the flora of liverworts in the Murmansk Region over the past 30–40 years has been paid to mountainous areas. Lowlands and, in particular, wetlands and forest areas, are much less studied. This is most likely due to the fact that no species of mires, such as *Rudolgaea borealis*, occurring in Scandinavia and also in Siberia, has been found in the region. It is difficult to explain such a strange gap in the distribution of this species with anything other than under-collecting.

### 3.3. Threatened Species

Of the 178 liverworts assessed in the European Red list of bryophytes against one of the threatened categories, 40 species occur in the Murmansk Region. Five species are classified in Europe as critically endangered (*Scapania sphaerifera* and *Calycularia laxa*) or endangered (*Cephaloziella integerrima, Schizophyllopsis sphenoloboides* and *Cephaloziella polystratosa*). All these liverworts are very rare and also endangered in the Murmansk Region. Fourteen species found in the Murmansk Region are assessed in the European Red List of bryophytes as vulnerable. The majority of these are rare and considered threatened in the region, e.g., *Isopaches decolorans, Lophoziopsis pellucida, Mannia triandra, etc.* Furthermore, some of the species are assessed as vulnerable in Europe due to the wide distribution of suitable habitats that are located in hard-to-reach areas. Specifically, Arctic alpine species restricted to late-snow areas (*Lophozia savicziae, Marsupella condensata, Moerckia blyttii)* are not considered threatened in the region. Arctic alpine calciphile *Mesoptychia gillmanii* is also not rare in the region. Of the species assessed as near threatened in Europe and found in the Murmansk Region, the majority are rather widespread and not threatened in the MR (*Geocalyx graveolens, Odontoschisma francisci*, etc.), though some are very rare in the region, e.g., *Aneura mirabilis.* Of the species classified in Europe as data deficient, eight occur in Murmansk Region. This group includes the recently described (*Frullania subarctica),* the poorly known (*Lophoziopsis rubrigemma, Lophozia silvicoloides),* as well as the taxonomically unclear (*Lophozia murmanica*) or potentially overlooked (*Barbilophozia rubescens, Isopaches alboviridis, Scapania obscura*) species. The majority of the red-listed European liverworts occur in the newly established National Park Khibiny, as well as in the Lapland State Nature Reserve and the Sunctaury (Zakaznik) Kutsa. Given this, as well as the uneven study of the Murmansk Region and the fact that many of the mountainous and lowland territories of the region have still been very poorly studied, it can be argued that most of the species, including those now considered rare, will be found in additional locations and the Murmansk Region can be considered as a refuge of many Arctic and North taiga species.

## 4. Materials and Methods

The annotated list of liverworts of the Murmansk Region is compiled based on a critical revision of about 150 publications and label data of herbarium specimens stored in herbaria of the Polar-Alpine Botanical Garden-Institute of the Kola Science Centre of RAS (KPABG) and Institute of North Industrial Ecology Problems of the Kola Science Centre of RAS (INEP) and accumulated in the information system L. (former CRIS) [63].

In the annotated list, we provide links to publications containing data on the localities or occurrence of species across nine floristic provinces of the Murmansk Region (Figure 1), applied in the previous publication [1]. In cases where the species was not reported for the province in publications, but where voucher specimens which supported the occurrence of species in this province were available in the herbaria (KPABG or INEP), the herbarium number or author’s number of one of the voucher specimens is provided. Publications in which the species is given for the area for the first time are highlighted. We also found it useful to provide the links to articles that published data on specimens from the Murmansk Region with nucleotide sequence data, as well as links to herbarium numbers of species vouchers sequenced and deposited in GenBank from the Murmansk Region, which had not previously been published. In total, 23 accessions were published in this study. In the checklist, we also showed the threatened status of the species in the Red Data Books of Europe, Russia, as well as in the Murmansk Region.

## 5. Conclusions

An analysis of the compiled list of liverworts of the Murmansk Region revealed a number of problems in studying the flora of liverworts of the region. As mentioned above, it turned out that some widespread species are not represented in the herbarium in several botanical provinces. The second, third and eighth biogeographic regions are the least studied. For example, the absence of *Sphenolobus saxicola* in provinces I-IV, of *Tritomaria quinquedentata* in the second province, and of *Harpanthus flotovianus* and *Jungermannia eucordifolia* in certain areas are most likely due to this factor. Obviously, it is necessary to pay attention to these gaps in knowledge.

Many liverwort taxa including those currently not accepted have been described from the Murmansk Region, and it is necessary to conduct a comprehensive study of type specimens and fresh material from type localities to confirm or correct the taxonomic status of the described taxa.

In recent decades, new species and varieties have been described from different mountainous and northern regions mainly based on an integrative approach. Some of these species have already been found in the Murmansk Region, e.g., *Apopellia* spp. [69], *Blepharostoma* spp. [89], and *Frullania subarctica* [96]. Some liverworts, e.g., *Nardia pacifica,* which was found in the Karelia Republic [151], or *Marsupella subemarginata,* which has been found in mountains of central Europe [147], may well be found in the Murmansk Region. The revision of taxonomically complex genera (*Cephaloziella, Lophozia, Nardia, Marsupella, Solenostoma, Jungermannia, Scapania*, etc.) and the inclusion of specimens from the Murmansk Region in integrative studies will undoubtedly lead to records of new species for the region.

For 82 species and one variety, data on one or more loci of chloroplast or nuclear DNA were obtained and published (Appendix A). These data used in taxonomic studies, on the one hand, confirm the validity of our taxonomic conclusions, and on the other hand, are the basis for further phylogeographic studies of liverworts of the Murmansk Region.

## Figures and Tables

**Figure 1 plants-14-01590-f001:**
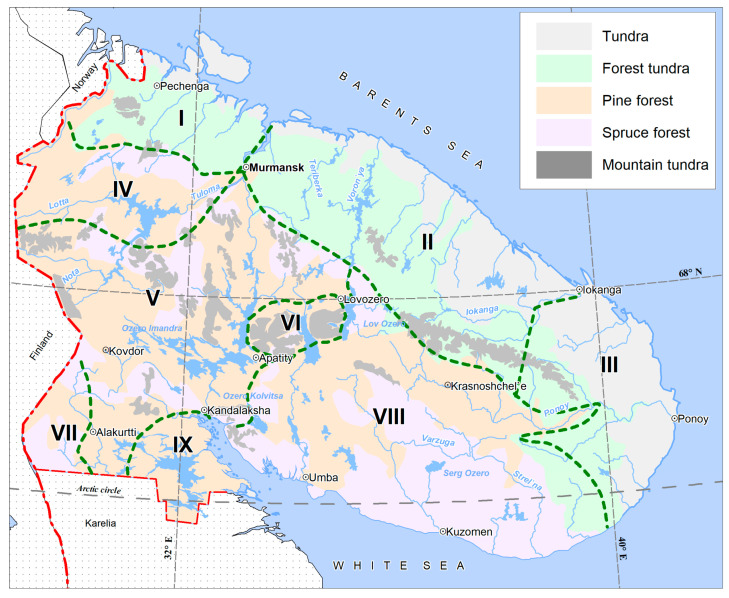
Floristic provinces of the Murmansk Region.

**Figure 2 plants-14-01590-f002:**
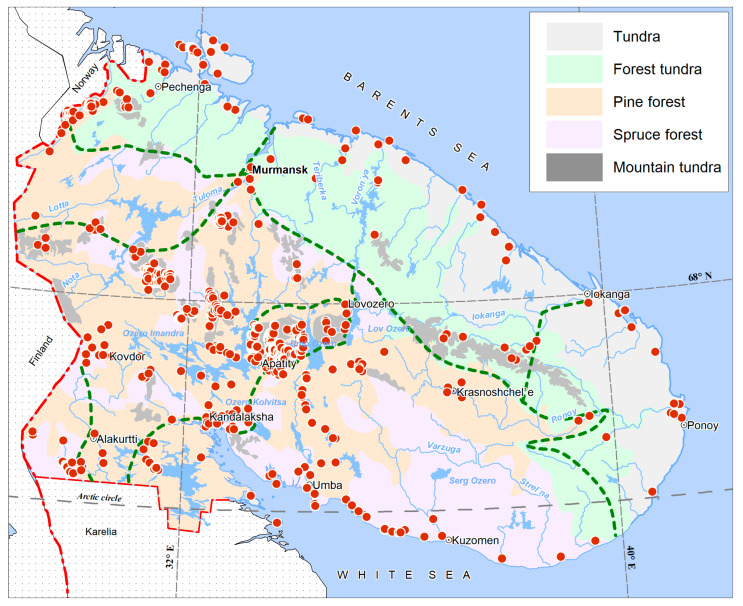
Collection points of liverworts in Murmansk Region.

## Data Availability

The raw data that support the findings of this study can be accessed in CRIS (L.)—Criptogamic Russian Informanion system, and are available at https://isling.org/ (accessed on 21 March 2025).

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
