# Peer review of "The Liverworts of the Murmansk Region (North-West Russia): Providing an Annotated Checklist as a Basis for the Monitoring and Further Study of Liverwort Flora"

_plants, 2025, doi:10.3390/plants14111590_

Round 1

Reviewer 1 Report

Comments and Suggestions for Authors The national and regional checklists represent an important synthesis
on the floristic consistency of a territory and the basis for further research
of both a floristic and ecological nature and suggests important ideas for the
conservation of species and a specific habitat.
We hope that the study will be extended in the near future also to Mosses
in order to have a complete picture on the consistency of the bryophytic flora of this territory.

Author Response

We thank the reviewer for appreciating our work. We agree that it is highly desirable to prepare a similar checklist of mosses. Such work is planned in the future.

Reviewer 2 Report

Comments and Suggestions for Authors

The review of work in the area is comprehensive, and it is very satisfying that concern for the history of field research is so evident. The correlation of station and climate/topography is excellent and quit interesting and will prove valuable in comparing the bryofloras of similar global sites. Of particular interest is the careful reference for particularly rare species in the area. The nomenclature if fully up to date, and the specimens are well-documented.

I could suggest that the details of the collections be placed in a data file online to save space, but given that these will be online anyway, there is no problem. It is best to keep all material together in one file, nowadays. 

The Comments section review problematic species and names, and make good judgments on correctness.

I agree that further work with type specimens and more sampling is necessary, but the paper is a thorough and intelligent place to start.  

I believe this paper summarizes the life work of the lead author, with additions by the co-author. Congratulations are extended!

I can think of nothing to change or emend in this paper. It should be published as it is.

Author Response

We thank the reviewer for appreciating our work. We agree that some of the specific data could have been provided in the application, but given that the publication is online, we found it more convenient to give it all together, which the reviewer also writes about.

Reviewer 3 Report

Comments and Suggestions for Authors

The list of species presented in the paper was prepared very carefully and in accordance with the announced methodology.
Each taxon is confirmed either by reference to published floristic data for the Murmansk Region, or by pointing to a specific herbarium specimen deposited in the KPABG or INEP. The authors clearly labeled new records for the region, provided specimen numbers, and reliably identified taxa previously erroneously noted. Synonyms have been properly included, and nomenclatural changes have been carefully noted. In a few cases, confirmation of occurrence is based on single collections or molecular data without full morphological elaboration, but these too are derived from material collected in the Murmansk area. The whole is consistent, transparent and in accordance with the best standards for documenting the flora of the region. There were no factual errors or instances of unjustified inclusion of species in the list.

Author Response

We thank the reviewer for appreciating our work.
